# Harnessing Setaria as a Model for C_4_ Plant Adaptation to Abiotic Stress

**DOI:** 10.3390/plants14243710

**Published:** 2025-12-05

**Authors:** Juan David Ferreira Gomes, João Marcos Fernandes-Esteves, João Travassos-Lins, Andres Felipe Gaona Acevedo, Tamires de Souza Rodrigues, Marcio Alves-Ferreira

**Affiliations:** Laboratório de Genética Molecular e Biotecnologia Vegetal, Universidade Federal do Rio de Janeiro (UFRJ), Rio de Janeiro 21941-853, RJ, Brazil; jdfgbio@gmail.com (J.D.F.G.); jmfe1999@gmail.com (J.M.F.-E.); jtlins@gmail.com (J.T.-L.); andresgaona80@gmail.com (A.F.G.A.); tatasr@gmail.com (T.d.S.R.)

**Keywords:** abiotic stress tolerance, *Setaria viridis*, *Setaria italica*, C_4_ photosynthesis, climate resilience

## Abstract

Climate change and the resulting abiotic stresses that emerge due to anthropogenic activities are the main causes of agricultural losses worldwide. Abiotic stresses such as water scarcity, extreme temperatures, high irradiance, saline soils, nutrient deprivation and heavy metal contamination compromise the development and productivity of crops on a global scale. In this scenario, understanding the response of C_4_ plants to different abiotic stresses is of utmost importance, as they constitute major pillars of the global economy. To further our understanding of the response of C_4_ monocots, *Setaria viridis* and *Setaria italica* have gradually emerged as powerful model species for elucidating the physiological, biochemical, and molecular mechanisms of plant adaptation to abiotic stresses. This review integrates recent findings on the morphophysiological, transcriptomic, and metabolic responses of *S. viridis* and *S. italica* to drought, elevated heat and light, saline soils, nutrient deficiencies and heavy metal contamination. Comparative analyses highlight conserved and divergent stress-response pathways between the domesticated *S. italica* and its wild progenitor *S. viridis*. Together, these findings reinforce *Setaria* as a versatile C_4_ model for unraveling mechanisms of abiotic stress tolerance and highlight its potential as a genetic resource for developing climate-resilient cereal and bioenergy crops.

## 1. Introduction

Climate change represents one of the most significant threats to the current global socioeconomic model, impacting various sectors, with a particular emphasis on agriculture. The increasing atmospheric concentration of carbon dioxide ([CO_2_]) has been directly associated with the rise in global average temperature, leading to a higher frequency and intensity of extreme climate events [1]. Given the continuous population growth and the resulting increase in demand for food, biofuels, and other agricultural products, implementing strategies to mitigate the effects of climate change on agriculture is crucial for global food security and socioeconomic stability [2,3]. Agriculture, in particular, is highly vulnerable to climate variability, facing critical challenges such as rising temperatures, altered precipitation patterns leading to droughts and floods, and an increasing frequency of extreme weather events projected for the coming decades [4].

According to the Intergovernmental Panel on Climate Change (IPCC) report, the global average temperature increased by 1.1 °C between the late nineteenth century and the beginning of the twenty-first century, with projections indicating an additional rise of 2.0 °C by the end of the 21st century [5]. Among the primary abiotic stresses affecting agricultural productivity, extreme temperatures and water deficit stand out, accounting for annual crop yield losses ranging from 51% to 82% worldwide [6]. Water availability is widely recognized as one of the most critical limiting factors for agricultural productivity, with projections indicating that, over the next five decades, water stress alone will restrict productivity on more than 50% of the planet’s arable land [7].

Due to the necessity to maintain crop productivity in a changing climate scenario, one of the most central themes in plant biology is the unveiling of plant adaptation against oxidative stress in response to abiotic factors. Hence, adaptive responses and molecular mechanisms underlying plant-environment interactions can provide crucial tools for the development of management strategies and genetic improvement in agricultural crops [8]. Plants face various abiotic stresses, such as drought, light, temperature, heavy metals, hypoxia or anoxia, nutrient deficiency or toxicity, UV light stress, and pesticides, which negatively impact normal plant growth and development [9]. and promote the accumulation of reactive oxygen species, ultimately reducing growth and yield [10]. Furthermore, photosynthetic machinery is also severely affected (Sharma et al., 2019), in which the chemical reactions mediated by photosystem I (PSI) and photosystem II (PSII), as well as the biosynthesis of chlorophyll undergo several changes [11].

Stress events mainly limit plant photosynthesis by reducing the CO_2_ assimilation via the Ribulose-1,5-bisphosphate carboxylase/oxygenase (Rubisco) pathway [11], triggering the production of a large amount of reactive oxygen species (ROS) in plant organelles such as plastids, peroxisomes, and mitochondria. ROS act as crucial signaling compounds in key cellular mechanisms that affect overall plant growth and development [12]. ROS molecules, such as H_2_O_2_, singlet oxygen, hydroxyl (OH^−^) and superoxide radical (O^2−^) [12], are produced in excess under stress conditions. To mitigate ROS, plants have antioxidant enzymes, which include detoxifying enzymes such as catalase (CAT), superoxide dismutase (SOD), ascorbate peroxidase (APX) and glutathione peroxidase (GPX) [13]. Plants have developed other protective mechanisms, such as photorespiration, antioxidant systems, and alternative and cyclic electron flow to prevent photosynthetic losses [14].

C_4_ plants have a specific photosynthetic apparatus, which is particularly responsive to abiotic stress. They use distinct physiological and biochemical mechanisms to deal with abiotic stressors, but their sophisticated photosynthetic system also has limitations. Abiotic stresses affect the photosynthetic mechanism by reducing stomatal conductance, causing oxidative stress and decreasing the activity of Ribulose-1,5-bisphosphate carboxylase/oxygenase (RUBISCO) [15]. Ref. [16] noticed they maintain greater photosynthetic efficiency despite moderate heat and water limitation due to their CO_2_-concentrating mechanism but show rapid decreases in assimilation when temperature or drought exceed their optimal thresholds. Under drought stress, C_4_ species respond with increased root-to-shoot ratio, osmolyte accumulation, and antioxidant defenses, although mesophyll-bundle sheath coordination frequently limits carbon fixation earlier than in many C_3_ species [17]. Heat stress tolerance has been connected to species-specific strategies for sustaining Rubisco activase function at high temperatures, emphasizing the significance of isoform variety in C_4_ grasses [18]. In response to salinity, chloroplasts emerge as primary targets, and tolerance is dependent on ion homeostasis, proteome changes, and the ability to mitigate oxidative damage [19]. These findings show that the resilience of C_4_ plants to abiotic stress needs to be deeply explored as a potential tool for plant breeding.

In this context, *Setaria viridis* and *Setaria italica*—also known as green foxtail and foxtail millet—appear as model plants for Panicoid grasses [20,21], especially because of the phylogenetic and metabolic proximity to species of economic interest of the Panicoideae family. In the case of *S. viridis*, it has a small diploid genome with a telomere-to-telomere chromosomal assembly of 395.1 Mb [22], as well as a short size (30 cm, on average), rapid life cycle (6 weeks, seed to seed), and large seed production, which made it a suitable model system for other C_4_ monocots [22,23]. For *S. italica*, the most recent genome assembly of the Yugu1 variety reports a total genome size of 405.7 Mb, which is similar to its wild counterpart. However, the size and life cycle differ greatly from *S. viridis*, since *S. italica* plants can reach up to 215 cm of height and have a generation time of 8–15 weeks from planting to seed maturity, depending on the accession [24]. Since its proposal as a model plant, the development of Agrobacterium-mediated transformation and CRISPR/Cas9 gene editing protocols contributed to its successful establishment as a model system [25,26,27].

It is widely considered that *S. italica* was domesticated in China from its wild ancestor *S. viridis* between 9000 and 6000 years before present time [28]. Following its domestication, *S. italica* became a widespread and staple crop in Chinese history, being referred to as one of the “Five Grains”, which were essential crops accounting for the majority of plant fossils in archaeological sites in China [29]. Currently, it is mostly cultivated in China and some parts of India, as well as in the USA, Japan, Indonesia, Australia, and other countries [30]. Its long history of cultivation in different countries resulted in a vast diversity of genetically and morphologically distinct landraces and cultivars, which have been employed by researchers to study the genetic basis of different plant traits [31]. In contrast to *S. italica*, which is a cultivated crop, *S. viridis* is a globally distributed invasive weed, that disturbs several plantations by competing for resources and hindering the growth of cereal crops [32]. In recent years, however, since its proposal as a model system for other C_4_ grasses, a myriad of studies have used it to study root development [33], response to abiotic stresses [34], biomass accumulation [35], among other traits. Though distinct in many aspects, both *Setaria* species share multiple similarities in their response to abiotic stresses, which resulted in the parallel emergence and adoption of both as model species for such studies.

Although plants have been extensively investigated for their physiological and molecular responses to abiotic stresses, a central question remains unclear: how do C_4_ plants respond to such stressors? In particular, the responses of *Setaria* species under conditions such as drought, heat, and salinity are still not fully understood. Recent studies suggest that *Setaria* exhibits species- and genotype-specific adaptations, including adjustments in photosynthetic efficiency, osmolyte accumulation, antioxidant activity, and gene expression related to stress tolerance [36,37,38]. In this review, we challenge ourselves to provide a comprehensive overview of current knowledge on the physiological, biochemical, and molecular mechanisms by which *Setaria* spp. associated with abiotic stresses, highlighting potential pathways for crop improvement and climate-resilient agriculture.

## 2. Drought Stress

Drought is probably the most well-studied abiotic stress in plants. And rightfully so, according to a report by the Food and Agriculture Organization of the United Nations, drought is responsible for 34% of the global losses in crop and livestock production [39]. Droughts are severely detrimental, since the high temperatures and reduced water availability compromise the plant metabolism, impair growth and reduce the quality and yield of crops [40] (Figure 1). For now, we will focus on the water deficit component of drought, seeing as high temperatures will be discussed in a further section.

The lack of water compromises the development of all plants, including monocots, and members of the *Setaria* genus are no exception. Prolonged exposure to conditions of water deficit stunts the overall growth of *S. italica* and *S. viridis*, as shown by the reduced shoot length [41], decreased dry weight of the shoots [34,42], and impaired emergence of leaves and tillers [43]. Another effect of the water deficit is a premature emergence of panicles, which indicates an early transition into the reproductive phase [33]. Moreover, the yield is also compromised by drought stress, since reductions in the number of panicles and in the weight of individual panicles and grains have been observed [41,44].

Even though most studies focus on the effects of drought in the leaves and shoots, it is of the utmost importance to evaluate it in roots, since they are the first structure to perceive the reduced water availability. In foxtail millet, withholding the watering for 8 days resulted in an increase in the total root length and surface area, which may be a strategy to maximize the water absorption [38]. In multiple accessions of *S. viridis*, on the other hand, considerable reductions in the root system have been described due to a suppression in the postemergence growth of crown roots [33]. In contrast to *S. viridis*, *S. italica* exhibited an ability to maintain the growth of a small number of crown roots during water stress. This difference likely originated during the domestication process of *S. italica*, since the arrest in crown root emergence seems to be a conserved response to drought among other Poaceae species, such as sorghum, maize and switchgrass [33].

Considering that the decrease in water absorption compromises the accumulation of water in the tissues, the severity of the water deficit is often evaluated by measuring the decrease in relative water content (RWC) of leaves. This approach has been effectively applied in studies with *S. italica* [38,42] as well as *S. viridis* [33]. Another metric used to measure plant water status is the leaf water potential (LWP), which has also been applied in foxtail millet [45] and green foxtail [34]. The two parameters have been used to differentiate between drought-tolerant and drought-sensitive genotypes in both species [34,41,46], indicating that they are well-suited for this type of screening.

To cope with the decreased water availability and reduce the water losses by transpiration, grasses often go through a process of leaf rolling [47,48]. In *S. italica*, leaf rolling and a reduction in the exposed leaf area have been reported in different cultivars, such as Yugu1, Jigu39, Jingu21, and Longgu16 [38,42,49]. In *S. viridis*, a reduction in the exposed leaf area has been observed after air-drying treatment [50], but not after a milder water deficit promoted by 7 h of polyethylene glycol (PEG) exposure [46]. Another visual sign of drought stress in species of Poaceae is the loss of pigmentation and turgor in the leaves, which have been described in *S. italica* [42,49] and *S. viridis* [43]. Additionally, drought-sensitive varieties often exhibit earlier leaf rolling, withering and bleaching, when compared to drought-tolerant ones after exposure to water deficit [41,45].

One other consequence of the water deficit is a decrease in photosynthetic activity. Since plants reduce stomatal aperture even under mild water-limiting conditions, photosynthesis is limited due to the decreased CO_2_ diffusion from the atmosphere to the carboxylation site [51]. Consequently, gas exchange measurements are often used to indirectly monitor the photosynthetic activity of stressed plants. In *S. viridis* and *S. italica*, the CO_2_ assimilation (A), transpiration (E) and stomatal conductance (gs) have been shown to decrease proportionally to the level of water deficit [42,52]. Gas exchange measurements, therefore, can be a powerful tool for the identification of drought-tolerant [42,52] accessions, as demonstrated in *S. italica* by [41]. In drought-sensitive accessions of green foxtail, such as Ast-1, the water deficit results in a more marked decrease in A, E and gs, when compared to drought-tolerant ones, such as A10.1 [34]. On the other hand, [52] found that Ast-1 showed a higher resistance to dehydration than A10.1 by maintaining higher levels of A during longer periods of water deficit, contrasting with previous observations [34,46]. The Water Use Efficiency (WUE), which is a ratio between CO_2_ A and E, can also be used to monitor the drought tolerance of plants, since it represents how much inorganic carbon is being assimilated for each water molecule lost by transpiration. By using an interspecific *S. viridis* × *S. italica* recombinant inbred line (RIL), [53] devised a modeling approach to predict the relationships between WUE and plant size, demonstrating that WUE exhibits high heritability. The study also illustrated that WUE is responsive to soil water availability, which was previously described. In C_4_ monocots, especially in drought-tolerant cultivars, WUE may increase under water-limiting conditions, as demonstrated in *S. viridis* and *S. italica* [45,52]. Another reason for the decrease in photosynthetic activity during drought is the degradation of pigments and other components of the electron transport chain in the thylakoid membrane.

The process of chlorophyll degradation was previously characterized in *Setaria viridis* [34,46] and *Setaria italica* [38,54]. In drought-sensitive cultivars of foxtail millet, this degradation occurs more intensively than in drought-tolerant cultivars, indicating a greater susceptibility of the photosynthetic apparatus to water deficit [49]. This is also observed in green foxtail, in which this degradation occurs concomitantly with higher repression in chlorophyll synthase gene expression [34], an enzyme involved in chlorophyll biosynthesis. In both species, a preferential degradation of chlorophyll a (Chl a) relative to chlorophyll b (Chl b) has been observed under water-limited conditions [46,55]. During drought stress, Chl a is preferentially degraded compared to Chl b due to its central role in the reaction centers of photosystems I and II, where it directly participates in the conversion of light energy into chemical energy. This functional position exposes Chl a to excess excitation energy and the formation of reactive oxygen species (ROS), which are enhanced by the reduction of CO_2_ assimilation caused by stomatal closure under water deficit conditions [56].

The degradation of components of the electron transport chain and the diminished CO_2_ uptake during drought reduce the available pool of oxidized electron acceptors (quinones and plastoquinones). Environmental stress also accelerates the photoinhibition of Photosystem II (PSII) [57]. Therefore, during drought, the non-photochemical dissipation of the chlorophyll excitation energy is increased, mainly in the form of heat and fluorescence, which can be monitored through the measurement of chlorophyll fluorescence kinetics. In *S. viridis* and *S. italica*, it is well documented that the limited water availability leads to a reduction in the maximum photochemical efficiency of PSII (Fv/Fm), as well as the effective photochemical efficiency (ΦPSII) [43,45,46]. It is worth mentioning that even though C_4_ plants exhibit a higher WUE than C_3_ plants and are generally considered more photosynthetically efficient, they often present lower Fv/Fm values [57]. In *S. italica* and *S. viridis*, this phenomenon is frequently accompanied by an increased non-photochemical quenching (NPQ), which represents the thermal dissipation of the excess energy, as well as a decrease in the coefficient of photochemical quenching (qP), which represents the proportion of open PSII reaction centers [42,50]. More specifically, in *S. viridis*, an exposure of 3 to 10 days also resulted in a decrease in electron transport quantum yield (ΦEo) and efficiency (ΨEo) [43]. The multiple chlorophyll fluorescence parameters can also be used to differentiate between drought-tolerant and drought-sensitive accessions and cultivars, as demonstrated in *S. italica* [45] and *S. viridis* [46].

Reductions in photochemical efficiency are often also attributed to photoinhibition due to damage caused by the increased amount of reactive oxygen species (ROS) during drought stress. There are several strategies to indirectly measure the oxidative damage, such as the quantification of the malondialdehyde (MDA) content, which is generated by ROS-induced lipid peroxidation. The increase in MDA content is well documented in several species and green foxtail, and foxtail millet are no exceptions [34,38,42,58]. Another strategy to evaluate the oxidative damage during drought stress, which has been effectively applied in leaves of *S. viridis* and *S. italica*, is the measurement of electrolyte leakage. The relative electrolyte leakage can be used to infer the severity of oxidative stress, because it is a consequence of the oxidative damage in cell membranes. Under distinct water deficit conditions, increased levels of electrolyte leakage have been observed in leaves of *S. viridis* and *S. italica* [38,45,46,50]. In roots, however, a reduction in electrolyte leakage levels was observed after 6 and 10 days of exposure to PEG-8000 (7.5%) [43].

To respond to the rising levels of ROS during water deficit, plants often increase the synthesis of ROS-scavenging enzymes, such as superoxide dismutase (SOD), catalase (CAT), peroxidase (POD), glutathione peroxidase (GPx), lipoxygenase (LOX), among others. In *S. italica* (Yugu1), exposure to drought conditions induced the expression of ascorbate peroxidase (APX) in seedlings [38]. In mature plants of the same cultivar, increased SOD and POD activities were detected, as well as an up-regulation of LOX1 and LOX5 [55], involved in stomatal closure, activation of antioxidant enzymes, and osmoprotectant synthesis, LOX1 and LOX5 help mitigate the accumulation of reactive oxygen species (ROS) and limit cellular damage under drought stress. Additionally, LOX1/LOX5-derived oxylipins contribute to jasmonic acid (JA) biosynthesis, regulating the expression of drought-responsive genes and enhancing plant survival and physiological maintenance [59,60]. Compared to Yugu1, the POD activity of the drought-sensitive *S. italica* variety AN04 was much lower [58]. In the tolerant *S. italica* M79 genotype, drought resulted in the up-regulation of 63 differentially expressed genes (DEGs) involved in redox regulation, such as genes encoding SOD, LOX, APX, GPx and POD [45]. This same genotype also exhibited increased CAT activity in relation to its parental genotypes. Similar results have been observed in the drought-tolerant Zha-1, A10.1 and Ula-1 accessions of *S. viridis*, which exhibited higher CAT expression and catalase activity after 7–10 days of water withholding when compared to drought-sensitive accessions [34]. Moreover, Amoah et al. (2023) proposed “drought hardening” as an efficient strategy to mitigate future drought stress [61]. Acclimated *S. italica* showed higher drought tolerance during subsequent stresses compared to non-acclimated plants. “Hardened” plants displayed enhanced antioxidant defenses and elevated anthocyanin and polyphenol content, resulting in more effective ROS scavenging and increased photosynthetic efficiency, total biomass, tissue water content and photosynthetic pigment content.

Another plant response to drought-induced oxidative stress is to increase the synthesis and accumulation of osmolytes and antioxidant metabolites. Among the metabolites that accumulate during water deficit, proline is probably the most well-studied. It is an osmoregulator, shielding cell molecules, organelles and membranes from the damage caused by ROS, but it also acts as a storage of carbon and nitrogen in stressed plants [62]. In *S. italica* and *S. viridis*, different methods to promote water deficit have been shown to induce the accumulation of proline in leaves [34,54]. The up-regulation by water stress has also been reported for genes of the proline biosynthesis pathway, such as pyrroline-5-carboxylate reductase (*P5CR*) and delta-1-pyrroline-5-carboxylate synthase 2 (*P5CS2*) [34,42,46]. In roots, there is less information available, but 6 and 10 days of exposure to PEG-8000 (7.5%) promoted only a slight induction in the expression of *P5CS2* in *S. viridis*, with no significant alteration in the proline content [43]. Along with proline, other metabolites that play a role in osmoregulation have been reported to accumulate during drought in foxtail millet and green foxtail [42]. Cultivars of both species that are more tolerant to drought have been shown to contain a higher content of soluble proteins and soluble sugars, which can help with osmoregulation [34,41,49]. Increased levels of glycinebetaine and gliadin have also been reported in *S. italica* under conditions of water deficit [44,54]. Glycinebetaine acts as an osmoprotectant, contributing to the stabilization of proteins and membranes, maintenance of enzymatic activity, and reduction in oxidative damage during dehydration stress. In contrast, gliadin is a storage protein whose accumulation may be altered under drought conditions, potentially affecting the protein composition and quality of the grains [44,54]. In the roots of *S. italica* (Yugu1), [38] reported greater alterations in the protein abundance during drought stress when compared to leaves of the same plants. Conversely, after rewatering, the protein abundance of leaves is mainly associated with photosynthetic activity, while the protein abundance of roots seems to be primarily related to the regulation of secondary metabolism.

Plants possess several mechanisms for transducing the stimulus of a water deficit into physiological and molecular responses. Phytohormones are one of the main components of drought signal transduction and, among them, abscisic acid (ABA) plays a major role in the regulation of signaling during desiccation. The ABA content has been shown to increase under water deficit conditions in *S. italica* [42]. In *S. viridis*, drought-sensitive accessions exhibited the highest ABA levels during drought, as well as the highest up-regulation of zeaxanthin epoxidase (ZEP) and 9-cis-epoxycarotenoid dioxygenase (NCED), which are involved in the biosynthesis of ABA [34]. The activation of ABA-responsive genes depends on a signaling cascade composed of three major components: ABA receptors of the PYR/PYL/RCAR family, 2C protein phosphatases (PP2Cs) and Snf1-related protein kinases 2 (SnRK2s) [63]. Recently, [64] characterized the ABA pathway signaling components in *S. viridis* and *S. italica* and found a high conservation in all three families. Besides one PP2C gene that was duplicated in *S. viridis*, all other genes exhibited a 1 to 1 orthology relationship between the two species. The main differences between the orthologues were due to the shorter 5′ and 3′ untranslated regions (UTRs) of the *S. italica* copies. Studies with both species report that PYL genes are downregulated by the water deficit [64,65], while PP2C genes have been reported as being upregulated [65,66] or downregulated [34], depending on the experimental conditions. The expression pattern of SnRK2 genes seems to show great variability in response to drought, but is mostly upregulated [34,64,65]. The ABA signaling cascade culminates in the activation by SnRK2 of several transcription factors, which in turn regulate the expression of multiple other genes involved in the drought response. In *S. italica*, the promoters of water deficit-induced DEGs are enriched with ABA-responsive elements (ABREs), which are binding sites for the ABF (ABA-responsive element binding factors) family of transcription factors [66]. The ABF family is also induced by drought stress in plants and *S. italica* is no exception [42]. Previous reports have also shown that promoters of ABA signaling components of *S. viridis* contain ABREs [65], which may indicate a positive feedback regulation of ABA response.

Besides ABFs, there are other transcription factor families that play a major role in the regulation of gene expression in response to drought. Among them, the MYB, NAC (NAM, ATAF1/2, and CUC2), DREB/CBF (Dehydration-responsive element binding/C-repeat binding factor), WRKY, bZIP, HD-ZIP and HSF (Heat Shock Factors) families stand out as the more well-known. Different RNA-seq projects with *S. italica* have demonstrated that transcription factors from the aforementioned families are differentially expressed by the water deficit [45,66,67]. Within these transcription factor families, reliable drought marker genes have been identified for *S. viridis*, such as NAC6 and DREB1C, both of which are up-regulated by PEG and air-drying-induced water deficit and seem to aid in the adaptation of the plant to drought [43,50]. It is worth mentioning that drought-marker genes which are repressed by water deficit have also been identified. One example is *WRKY1*, which negatively regulates drought response by repressing the expression of MYB2 and DREB1A, as well as inhibiting the ABA-mediated stomatal closure [68]. In *S. viridis*, *WRKY1* is repressed after 7 h of PEG-8000 exposure in the drought-tolerant A10.1 accession but is induced in the sensitive Ast-1 accession [46]. Besides the previously mentioned transcription factor families, transcriptomic studies have also identified DEGs from other less discussed families, such as bHLH, DOF, C2H2 and ERF [49,67]. Recently, [69] conducted a characterization of the MADS-Box family of transcription factors in foxtail millet and green foxtail, which is not often associated with drought response. Nonetheless, a considerably large number of cis-elements associated with dehydration response were found in the promoters of *SiMADS-Box* genes. Moreover, overexpression of *SiMADS51* in Arabidopsis resulted in a lower drought stress tolerance, as evidenced by the impaired germination, shorter roots, reduced fresh weight and lower survival rate under water deficit conditions. In rice, overexpression of *SiMADS51* also resulted in a lower survival rate after drought stress, along with reduced POD and SOD activity [69]. The authors propose that *SiMADS51* may act as a negative regulator of the drought stress response, inhibiting the expression of stress-related genes involved in stress response (*DREB2A, MYB2*), ABA biosynthesis (*NCED1, NCED3*), ABA signaling (*PP2C06, PP2C49*) and proline synthesis (*P5CS1*).

Although transcription factors are often the focus of studies about gene expression regulation, a growing field is the regulation by non-coding RNAs (ncRNAs), such as long non-coding RNAs (lncRNA), small interfering RNAs (siRNAs) and micro RNAs (miRNAs). Though scarce in the literature, some studies have tried to characterize this regulation in species of the *Setaria* genus exposed to water deficit. A study by [66] identified differentially expressed siRNAs and lncRNAs in foxtail millet seedlings treated with PEG-6000 for 7 h. Few lncRNAs were identified as regulated by the water stress, but the clusters of siRNAs of 21 nt and 24 nt were enriched in specific gene-rich regions of the genome, which suggests a role in transcription regulation during drought response. A more recent study with *S. viridis* performed small RNA deep sequencing to identify differentially expressed miRNAs and their targets in response to water deficit conditions [52]. Among the identified miRNAs, some of them targeted transcription factors from the MADS-Box, MYB, and NAC families. Curiously, several novel miRNAs targeted genes involved in cell wall synthesis and remodeling, especially during the early responses to drought, which may be an initial adaptive mechanism to water deprivation.

During drought stress, ncRNAs and transcription factors regulate gene activation and repression, shaping the plant’s transcriptional response. RT-qPCR is commonly used to assess these expression changes, requiring stable reference genes for accurate normalization. In *S. italica*, RNA POL II and EF-1α were identified as reliable reference genes under drought, with APRT also showing stable expression [70]. In *S. viridis*, suitable reference genes include SDH, KIN, and SUI1 [71]. Additionally, [50] recommended using TPI + GAPDH during the vegetative stage and Cullin + Ubiquitin-conjugating enzyme E2 during the reproductive stage for normalization in drought experiments.

There are multiple classes of genes that are regulated in response to the drought stimulus. In general, photosynthetic components are down-regulated during water stress, as seen in *S. viridis* in a study by [34], in which multiple photosynthesis-related genes were repressed, especially in the drought-sensitive Ast-1 accession. The list of repressed genes included photosystem 1 (PS1), ribulose bisphosphate carboxylase small subunit (RbcS), phosphoenolpyruvate carboxylase (PEPC), among others. Similar results have been observed in *S. italica*, since water deficit resulted in the downregulation of photosystem II, photosystem I, and cytochrome b6/f complex-related genes [49]. Furthermore, RNA-seq experiments with foxtail millet have shown that the downregulation of photosynthesis-related genes during water deficit is often accompanied by a repression of carbohydrate metabolism genes [49,66].

There is also a myriad of classes of genes that are normally up-regulated and several of them show great promise for biotechnological applications. Heat-shock proteins are one example, since they act as molecular chaperones, ensuring correct folding of other proteins and have been shown to be up-regulated by water deficit in both foxtail millet [41,66] and green foxtail [43]. Late embryogenesis abundant (LEA) proteins are another notable example, due to their role in protecting cellular components from water-stress-induced damage, assisting with protein folding and acting as molecular chaperones for other proteins [72]. Among them, the group II of LEA proteins, which are known as dehydrins, are of particular interest, since their overexpression has been shown to confer tolerance to multiple abiotic stresses [73]. In RNA-seq data of *S. italica*, up-regulation of several LEA proteins and dehydrins has been recorded following water deficit [45,66], while in *S. viridis SvDHN1* and *SvLEA* have been shown to be induced by PEG in water stress conditions [43,50]. Another group of proteins with significant biotechnological potential is the membrane water channels, also known as aquaporins. In *S. italica*, aquaporin genes are often up-regulated by drought stress as a mechanism to increase water uptake and transport [45,66]. Furthermore, in a study by [34], drought-sensitive accessions of *S. viridis* exhibited the lowest expression of aquaporins PIP-1, PIP1-2 and PIP2-1 genes when compared to drought-tolerant accessions.

The fact that *S. italica* is a species domesticated from its wild relative *S. viridis* allows for the use of interspecific recombinant inbred lines (RILs) to study different aspects of the drought response. As mentioned previously, [53] applied this strategy to identify Quantitative Trait Loci (QTL) associated with water use efficiency. By doing so, their results showed that alleles from both parental species contribute to the WUE, indicating that neither is fully optimized. This approach has also been applied by Qie et al. (2014) [74] to study the genetic basis of drought tolerance, using a RIL population generated from a cross between the *S. italica* cultivar Yugu1 and the *S. viridis* W53 accession. Their results identified 18 QTLs, 8 of which were alleles from *S. viridis* that contributed to drought tolerance. A *S. viridis* × *S. italica* RIL population has also been used to investigate in grass species the response of the crown roots to water loss [33]. Through QTL mapping, the authors showed that the responses of crown roots to water deficit are regulated by a small number of specific loci. These loci were probably selected during the domestication process of *S. italica* and likely contribute to the higher tolerance to desiccation of its crown root system when compared to *S. viridis*. This approach based on the comparison of wild and domesticated cultivars has been applied to great success, as shown in a recent study by [75], which analyzed the *SiUBC39* gene. Their results indicated that the *SiUBC39* gene was strongly subjected to selection during the domestication of *S. italica* and, by evaluating CRISPR knockout lines, they were able to obtain phenotypes similar to *S. viridis* (reduced plant height and grain weight). Moreover, the mutant plants exhibited a superior performance under drought stress, highlighting the potential of such comparative approaches for the discovery of new targets for genetic improvement [75].

## 3. Extreme Temperatures

Anthropogenic climate change has increased the frequency of extreme temperature events, such as heat and cold waves, leading to significant losses in agricultural productivity and threatening global food security [76]. Temperatures outside the optimal range disrupt cellular homeostasis, impairing plant growth, development, and metabolism [77] (Figure 2). Among temperature extremes, heat stress has received greater scientific attention than cold stress, and studies on chilling in *Setaria* species remain limited, despite its potential impact on agronomically important crops. Under cold stress, *S. viridis* exhibits repression of *SvPYL* and induction of *SvSnRK2* and *SvPP2C*, alongside marked reductions in photosynthetic efficiency and changes in stomatal conductance [65]. Furthermore, [78] reported extensive transcriptomic reprogramming (7911 DEGs), activation of osmoprotective pathways, cellular remodeling and signaling, repression of core biological processes, and the prominence of key regulators such as CBF-L, TINY1/2, AP2/ERF, MYB, and BBX under cold stress [78]. Another recent study demonstrated how *S. viridis* accessions A10.1 and Ast-1 under gradual or sudden cold stress suffer significant reductions in gas exchange rates and total biomass, as well as damage to the photosynthetic machinery [79]. *S. viridis* is also known for its high thermal plasticity, with optimal performance between 29/19 °C and 35/25 °C, and reduced growth under extreme temperatures, reinforcing its physiological and morphological adaptability [80].

Conversely, high temperatures are an increasingly critical threat, compromising essential functions such as photosynthesis, membrane integrity, protein stability, and hormonal balance. A detailed understanding of the morphophysiological, chemical, and molecular responses of plants to heat stress is fundamental for developing biotechnological strategies that enhance thermal productivity [81]. The identification of metabolic pathways and key genes involved in heat adaptation is crucial for the development of cultivars that are more resilient to extreme climatic conditions [82].

Plants of *S. viridis* subjected to 42 °C/32 °C (day/night) showed a 50% reduction in dry biomass of roots and shoots, with no change in the root/shoot ratio. The typical phenotype includes pronounced dwarfism and atrophy, though hormonal analyses indicated that the levels of salicylic acid (SA), jasmonic acid (JA), indole-3-acetic acid (IAA), and phenylacetic acid (PAA), an auxin analogue, remained unchanged under these conditions [83].

A comparison between plants under control conditions and those exposed to high temperatures revealed a reduction in dry weight during the flowering and grain-filling stages. This reduction is attributed to factors such as decreased height, reduced leaf area, and shortened growth period. Elevated temperatures directly impact leaf area, with significant reductions compromising the photosynthetic rate and, consequently, productivity. The strong correlation between reduced leaf area and leaf narrowing reflects an adaptive defense strategy against photo-oxidative damage. This mechanism is also observed in other important crops, such as maize [84], rice [85], and sorghum [86].

RuBisCO, the primary enzyme responsible for CO_2_ fixation in plants, has its activation limited under high-temperature conditions, compromising the balance of its inactivation [87]. This enzyme exhibits complex activities with variable kinetics in response to temperature, catalyzing the first steps in the photosynthetic and photorespiratory pathways, with reaction rates determined by carboxylase and oxygenase activities, which increase with rising temperature [88]. When CO_2_ fixation is inhibited at high temperatures, thylakoid energization is affected, as evidenced by changes in electrochromic absorption, non-photochemical quenching, and light scattering, indicating that the energy that would be used in the Calvin cycle is not absorbed [89]. The reduced RuBisCO activation at high temperatures is associated with its thermolability [90]. Hence, the reduction in RuBisCO levels is identified as a key factor determining the negative impacts of heat stress on photosynthesis [15]. However, plants possess certain plasticity to adjust photosynthesis to optimal growth temperatures, which includes changes in the ideal temperature for photosynthesis in response to seasonal variations, consequently enhancing the efficiency of the process under the new thermal conditions [91]. In *S. viridis*, RuBisCO exhibits kinetic responses to elevated temperatures that are comparable to those observed in C_3_ plant species, suggesting the evolutionary conservation of the enzyme’s kinetic parameters across species with different photosynthetic types [92] and its function.

ATP synthesis under moderate heat stress primarily occurs due to RuBisCO activation and photorespiration [18]. Moderate heat significantly affects the reactions of the cytochrome complex and PSI, which is more susceptible to damage than PSII under heat stress, while PSII and the stroma undergo oxidation, indicating a redox imbalance in the different compartments of the photosynthetic electron transport system [93]. The ability of plants to maintain optimal rates of CO_2_ assimilation and leaf gas exchange is directly proportional to their heat tolerance. Stomatal conductance and transpiration rate are closely related to leaf temperature, with the maintenance of net CO_2_ assimilation rates acting as a reliable indicator of the plant’s ability to tolerate high temperatures [94]. Temperatures above the ideal levels affect stomatal conductance, intracellular CO_2_ concentration, and leaf water status. Stomatal closure alters intracellular CO_2_ concentration under heat stress conditions, triggering the inhibition of net photosynthesis [95]. Furthermore, temperature changes directly influence the vapor pressure deficit (VPD), which modifies the plant’s hydraulic conductance and water supply to the leaves [96,97]. Chlorophyll biosynthesis in plastids is significantly compromised under heat stress, resulting in the degradation of chlorophyll molecules [98]. At elevated temperatures (35–45 °C), there is induction of cyclic electron transport and leakage from thylakoid membranes, compromising the integrity of the photosynthetic machinery [99]. Moreover, transcripts of key photorespiratory enzymes, including PGLP1, GOX, GGT1, SGAT, HPR, GDC subunits, pMDH2, Fd-GOGAT, and GS2, were markedly reduced. However, the proteins SGAT, SHMT, and HPR showed increased accumulation. This imbalance led to a decrease in serine and accumulation of glycerate, whose conversion to 3-PGA may be impaired due to the absence of GLYK protein. Additionally, glutamate and 2-oxoglutarate, both involved in ammonium assimilation, were significantly reduced [83]. Mild heat stresses have less detrimental impacts, whereas severe temperatures can cause irreversible damage. Nevertheless, cyclic electron flow at high temperatures maintains the energy gradient across the thylakoid membrane, preserving ATP homeostasis and preventing significant structural damage [100]. This energetic stability is crucial, as it allows photosystem I (PSI) and II (PSII), the CO_2_ reduction pathways, photosynthetic pigments, and electron transport systems to function effectively, ensuring the overall integrity of the photosynthetic machinery.

Together, photosystem I (PSI) and II (PSII), the CO_2_ reduction pathways, photosynthetic pigments, and electron transport systems are essential components of the photosynthetic machinery. Any deficiency in these elements results in a global inhibition of photosynthesis [101]. The photosynthetic apparatus acts as an environmental stress sensor, responding to cellular energy imbalances associated with modifications in the redox state. Among the responses to heat stress, photosynthesis is one of the most sensitive processes, with PSII being the first point of inhibition compared to other cellular structures [94]. This heightened susceptibility is mainly due to two factors: (i) the increased fluidity of thylakoid membranes, which displaces the light-harvesting complex, and (ii) PSII’s reliance on the dynamic integrity of electron flow.

PSII is especially susceptible to heat stress due to two main factors: (i) increased fluidity of thylakoid membranes, which displaces the light-harvesting complex, and (ii) PSII’s dependence on the dynamic integrity of electron flow. High temperatures may causecan lead to the dissociation of the water-oxidizing complex, displacement of the light-harvesting complex, and destabilization of the PSII reaction center [102].

Heat also induces the dissociation of ions such as Cl^−^, Mn^2+^, and Ca^2+^ from the PSII pigment-protein complex, as well as the release of extrinsic polypeptides from thylakoid membranes, further compromising the structure and functionality of PSII [103]. Among the intrinsic proteins of PSII, the D1 protein is particularly sensitive, being cleaved by the FtsH protease, which migrates from the stroma to the grana for degradation [104,105]. The diffusion of photodamaged D1 proteins is hindered by the loss of membrane integrity at extreme temperatures, reducing repair efficiency. Genetic studies suggest that manipulating the saturation levels of fatty acids in thylakoid lipids may increase resilience by initiating more efficient repair processes [106]. Under heat stress, transcripts of LHC II and PS II were downregulated, accompanied by significant reductions in the psbO, psbP, and psbQ proteins. In the Cytochrome b6f complex, all transcripts were markedly reduced, along with decreased levels of the PETA and PETC proteins [83]. In contrast, PSI exhibits greater thermal stability compared to PSII. Under high temperatures, the cyclic electron flow around PSI is intensified, contributing to the maintenance of the proton gradient in the thylakoids and promoting ATP production as a defensive mechanism [107]. Thus, PSI serves as a central element in protecting photosynthetic machinery from heat stress-induced damage.

LHC I and PS I subunits also showed reduced transcript levels, although with minimal changes at the protein level. In ATP synthase, transcript levels uniformly decreased, while ATPB, ATPC, ATPF, and ATPX proteins were significantly reduced. The NDH complex and PGR5, both associated with cyclic electron flow, exhibited strong reductions in both transcript and protein levels. This coordinated downregulation suggests a shared regulatory mechanism. Nevertheless, the electron transport capacity was maintained [83].

Exposure of *S. viridis* to high temperatures and light intensities results in significant reductions in net CO_2_ assimilation rates, with high light intensity inducing pronounced photoinhibition in the leaves [108]. During a 4 h treatment at 40 °C, leaf temperature increased from 31 °C to 37 °C [109]. This thermal increase directly impacted photosynthetic parameters, including PSII operating efficiency, stomatal conductance, transpiration rate, and electron transport rate, as assessed by gas exchange measurements and chlorophyll fluorescence [110]. Additionally, differential regulation of genes associated with metabolic pathways related to photosynthesis was observed, along with structural changes in chloroplasts.

Under heat stress, a reduction in the expression of several genes was noted, while the levels of SGAT [111], SHMT [112], and HPR [113] were increased. These genes are upregulated to support the photorespiratory cycle. SGAT facilitates the conversion of serine and glyoxylate to glycine, SHMT recycles glycine into serine while providing one-carbon units for metabolism, and HPR reduces hydroxypyruvate to glycolate [108,113,114,115,116]. Together, they help maintain carbon metabolism, prevent accumulation of toxic intermediates, and protect cells from oxidative damage, acting as a key adaptive response to heat stress. Metabolically, there was a decrease in serine and an increase in glycerate, which requires conversion to 3-PGA. However, the absence of detectable GLYK protein raises uncertainty regarding the relationship between glycerate accumulation and its metabolism. Other metabolites, such as glutamate and 2-oxoglutarate, also showed reduced levels in heat-stressed plants [86].

Plants of *S. viridis* under high temperatures exhibited a strong reduction in leaf starch accumulation, while sucrose, glucose, fructose, and various osmoprotective sugars accumulated intensely. Transcripts associated with starch biosynthesis (APL1, APS1, SS, and SBE) were downregulated. In the sucrose pathway, protein levels of cFBA, cPGI, cPGM, UGP2, and SPP were upregulated. Conversely, genes involved in the raffinose pathway were strongly induced, consistent with the pronounced accumulation of raffinose and galactinol, showing differential distribution between mesophyll and bundle sheath cells [83].

## 4. Light Stress

Light is a critical environmental signal that modulates photosynthesis, carbon assimilation, and overall plant growth. However, fluctuations in light intensity constitute significant abiotic stress, impacting a plant’s primary metabolism by disrupting various physiological, biochemical, and molecular processes [117] (Figure 2). Light stress occurs when the absorption of light energy exceeds the capacity for its use in photosynthesis. This over-excitation at the photosystems leads to photoinhibition, a process characterized by the functional failure of PSII reaction centers and a decline in photochemical efficiency [118,119]. A key mechanism of damage involves the highly oxidizing potential within PSII, which damages core proteins like D1. When the rate of D1 degradation surpasses its repair, PSII centers become inactivated [120].

A consequence of excess light is the generation of reactive oxygen species (ROS), which can damage both PSI and PSII, reduce mitochondrial activity, and force plants to dissipate excess energy as heat or fluorescence [15,121]. This negatively affects key photosynthetic parameters, including the maximum quantum efficiency of PSII, electron transport rate, and the chlorophyll/carotenoid ratio [122]. To mitigate the excessive ROS generation, plants employ antioxidant enzymes, protective compounds and repair mechanisms [123,124]. Specific protective compounds include plastoquinone-9, which acts as an antioxidant to reduce PSII photoinhibition [125], and secondary metabolites like anthocyanins [126]. Conversely, low light stress also impairs photosynthesis, primarily by reducing stomatal conductance and disrupting the photosynthetic mechanism. This leads to a dramatic increase in intercellular CO_2_ concentration and a decline in the net photosynthetic rate, transpiration rate, and water-use efficiency [127,128,129].

Ref. [130] evaluated the changes in carbon metabolism and in the transcriptome of *S. viridis* leaves acclimatized to high (1000 µmol m^−2^ s^−1^), medium (500 µmol m^−2^ s^−1^) and low light intensity (50 µmol m^−2^ s^−1^). Under low light conditions, photosynthetic efficiency is substantially impaired, leading to reduced growth, lower turgor, and diminished photosynthetic capacity. This is coupled with a significant reduction in key signaling sugars, namely glucose, sucrose, and trehalose-6-phosphate (T6P), which are crucial for downstream metabolic regulation. Conversely, under high light intensity, the photosynthetic machinery remains robust despite a marked accumulation of sugars. The induction of sugar accumulation under high light does not suppress photosynthesis. In fact, it suggests that *S. viridis* employs protective and compensatory mechanisms to mitigate any potential photoinhibitory effects. While low light primarily triggers a decline in the energetic and metabolic status of the plant, high light intensity drives a reallocation of carbon resources that may facilitate enhanced stress resilience. The expression of sugar signaling components is closely intertwined with light-mediated gene regulation in *S. viridis*. Low light caused a pronounced down-regulation of anabolic gene expression and an up-regulation of genes involved in catabolic processes, suggesting a metabolic shift toward energy conservation [130]. Central to this response is the modulation of hexokinase (HXK) and the sucrose non-fermenting 1 (Snf1)-related protein kinase, SnRK1. Under low light, the depletion of sugars results in the activation of SnRK1 targets in an attempt to reestablish energy homeostasis. In contrast, the sugar accumulation resulting from high light intensity represses SnRK1 signaling pathways, suggesting sugar availability may buffer the regulatory impact on energy-sensing mechanisms. Moreover, the differential expression of HXK under varying light conditions underscores the role of glucose sensing in fine-tuning metabolic processes based on light intensity [130].

While light intensity is a major determinant of photosynthetic performance and sugar metabolism, its impact is further modulated by interactions with other abiotic stresses, such as heat. Ref. [108] investigated how *S. viridis* responded to a four-hour period of high light and temperature. Their findings indicate that both stresses result in comparable reductions in photosynthetic efficiency. Transcriptomic analysis revealed key differences in differentially expressed genes between mesophyll and bundle sheath cells. Under high light, differentially expressed ROS-scavenging genes and HSPs were upregulated in mesophyll cells, while the bundle sheath-specific DEGs were downregulated. Curiously, the inverse was observed for ROS-scavenging genes under high temperature, while HSPs were upregulated in both cell types. The differential responses observed in mesophyll versus bundle sheath cells warrant further investigation to understand how cellular compartmentalization contributes to the overall resilience of C_4_ photosynthetic systems.

Comparative studies in *S. italica*, the domesticated foxtail millet, provide additional insight into how light stress influences photosynthesis and yield determinants. A field study conducted during the grain-filling stage demonstrated that increased shading leads to a marked reduction in net photosynthetic rate, stomatal conductance, effective quantum yield of PSII, and electron transport rate [131]. Conversely, intercellular CO_2_ concentration increased, reflecting a shift in the balance between CO_2_ supply and assimilation. Additionally, low light intensity changed the double-peak diurnal variation in photosynthesis to a single-peak curve, signifying altered light absorption and energy conversion processes [131]. These changes not only diminished the available assimilates for grain filling but also directly impacted yield by reducing fresh grain mass per panicle [131]. The sensitive response of *S. italica* to low light thus parallels the metabolic constraints observed in *S. viridis*, although the outcomes are more directly measurable in terms of agricultural productivity.

## 5. Salt Stress

Saline soils are characterized by the presence of water-soluble salts, with most studies on salinity stress focusing on sodium chloride (NaCl) due to its non-nutritional nature for plants [132]. Salt stress in plants can be split into two major components: osmotic stress, resulting from decreased soil osmotic potential, and ionic stress, caused by the excessive uptake of Na^+^ and Cl^−^ ions [133] (Figure 3). Soil salinization arises from natural processes, such as mineral weathering, or anthropogenic activities, with agricultural irrigation being the largest perpetrator [132]. Given the increasing challenges imposed by climate change on agriculture, and the rising need for artificial irrigation, soil salinization is becoming increasingly important and demands more studies to find sustainable alternatives when facing this problem.

Part of the ionic stress imposed by soil salinity is due to competition for ionic channels. Na^+^ competes with essential cations, such as Ca^2+^, K^+^ and NH^4+^, while Cl^−^ competes with anions like NO^3−^, potentially resulting in a nutritional deficit [132]. Ionic toxicity further inhibits photosynthesis, as evidenced by reductions in carbon assimilation and photosystem II (PSII) activity [134,135]. Evidently, as salinity increases, so do its adverse effects on germination, morphology and biomass accumulation.

Recently, several studies have explored the effects of salinity on *Setaria* species, ranging from its morphophysiological effects to the molecular mechanisms underlying salt stress. Seed germination, a pivotal stage in a plant’s life cycle, is particularly vulnerable to salt stress. Salinity commonly reduces germination rates, primarily through the modulation of ABA and ethylene levels, both of which regulate seed dormancy [136,137]. The negative effects of increased salinity on the germination rates of *S. viridis* and *S. italica* have also been explored. In *S. viridis*, elevated salinity delays germination, with higher NaCl concentrations significantly reducing germination rates. However, low salt concentrations appear to have minimal impact on seed germination [138,139]. Moreover, the effects of salinity on the germination of *S. italica* seeds are comparable to those in *S. viridis*. Recent studies have shown that a degree of variation in tolerance to salt exists among accessions [140,141]. Han et al. (2022) evaluated 104 *S. italica* accessions under 170 mM NaCl and reported a reduction in the germination rates, as well as in plumule and radicle length, underscoring the importance of genetic diversity in salt tolerance. Ref. [139] also demonstrated that seedlings of the A10.1 accession 9 days after sowing (DAS) suffered a large reduction in foliar area when grown in media containing 90 mM NaCl.

Salinity further impairs post-germination growth by reducing water uptake, inducing stomatal closure, and inhibiting carbon assimilation [133,142]. Recent studies on *S. viridis* revealed that salt stress significantly diminishes biomass accumulation [139,143]. Ref. [143] investigated the long-term effects of salinity on three *S. viridis* accessions, A10.1, ME034V, and Ast-1, and observed varying degrees of tolerance, with Ast-1 consistently presenting itself as the most sensitive of the three. While all accessions exhibited reduced total dry weight, Ast-1 showed a drastic decrease in shoot-to-root ratio, unlike A10.1 and ME034V, where this ratio increased [143]. Ref. [138] further reported substantial reductions in root and coleoptile length when seedlings were grown on media supplemented with 50 mM NaCl, with more severe effects at 100 mM.

In addition to reductions in biomass, salt stress induces distinct morphological changes, such as lesions on the primary and crown roots, as well as yellowing and swelling of the roots. Root emergence also seems to be impacted, as the final number and length of the crown roots are often reduced [143]. Aerial tissues are also affected, with symptoms such as leaf curling, burnt margins, and chlorosis [143]. Moderate salt stress causes flattening of the epidermal cells, while severe stress results in increased leaf thickness and intercellular spaces [144]. Severe stress may completely halt growth and cause leaf necrosis [143]. Ref. [139] further observed complete plant mortality within five days of irrigating *S. viridis* with solutions exceeding 25.8 dS/m salinity.

Similarly to drought, high soil salinity induces water stress by reducing soil osmotic potential [142]. In addition, the absorbed Na^+^ and Cl^−^ ions increase sap osmotic pressure, affecting water availability within leaf tissues [132]. This reduction in tissue water content leads to stomatal closure, a hallmark response to water stress in plants [11]. Stomatal closure in *S. viridis* under salt stress is associated with reduced CO_2_ assimilation rates [139,145]. Ref. [145] noticed that *S. viridis* exposed to 100 mM NaCl failed to recover from the stress under elevated CO_2_ conditions, highlighting the severe osmotic impact on carbon assimilation.

Given its detrimental effects on water balance and carbon assimilation, salinity also impairs photosynthesis. Abiotic stresses typically reduce photosynthetic efficiency by disrupting gas exchange, pigment biosynthesis, and the electron transport chain [11]. Chlorophyll fluorescence, a key indicator of photosynthetic performance [146], shows significant declines in PSII activity under salt stress. For instance, *S. viridis* exhibits reductions in photosynthetic rate (A) ranging from 30% (96 h under 100 mM NaCl) to 60% (200 mM NaCl) [65]. Similarly, declines in ϕPSII, Fv/Fm, and Fm are often accompanied by increases in initial fluorescence (Fo) and NPQ under salt stress [139,145].

Ref. [145] have also investigated the electron transport chain dynamics of *S. viridis* under salt stress, and provided evidence of photoinhibitory damage in PSII, even at low salt concentrations (50 mM NaCl). Under more severe and prolonged stress, a higher proportion of oxidized PSI reaction centers (P700) was reported, along with increased amounts of active P700. This finding suggests a possible mechanism to cope with adverse environmental conditions by maintaining PSI activity [145]. However, this photoinhibitory damage led to a notable reduction in the electron transport chain conductance, reflecting an overall decline in photosynthetic efficiency under high salinity stress [145].

In addition to structural damage, salt stress exacerbates the production of ROS, which are a common byproduct of abiotic stresses, including salinity [11]. In a separate study, [147] demonstrated that moderate salt stress (50 mM NaCl) promoted ROS accumulation in *S. viridis* due to the reduction in molecular oxygen within the chloroplasts.

Salt stress also induces significant alterations to chloroplast ultrastructure in *S. viridis*. Ref. [145] reported that even moderate salt stress led to deformation of bundle sheath cells, where chloroplasts became compressed and elongated due to the expansion of vacuoles. Moreover, the appearance of blank spaces within bundle sheath cells suggested intracellular salt deposition. Inside the chloroplasts, salt stress caused starch grains to increase in size and number while simultaneously reducing the number of thylakoid membranes [145]. Ref. [144] further documented a reduction in chloroplast numbers under severe salt stress conditions, emphasizing the profound structural impact of salinity on chloroplasts.

Abiotic stress, such as salt stress, threatens plant survival by disrupting cellular homeostasis, and to counter these adverse conditions, plants use multiple defense mechanisms, including phytohormone regulation [148]. Phytohormones, particularly ABA, play a central role in coordinating stress responses, including the activation of stress-responsive genes and the regulation of stomatal guard cells to maintain water balance [65,148]. Despite its importance, studies investigating the roles of phytohormones in *S. viridis* and *S. italica* under salt stress remain scarce, representing a significant gap in current knowledge.

Ref. [65] explored ABA signaling pathways in *S. viridis* under various stress conditions, including salinity. Their findings revealed that ABA levels increased in leaves of the accessions A10.1 and Ast-1 following salt stress treatments. Notably, Ast-1 plants exhibited a two-fold increase in ABA levels compared to A10.1 under high salinity. Interestingly, exogenous ABA treatments highlighted differences in sensitivity between the accessions: while carbon assimilation in A10.1 plants was affected by a 100 µM ABA dose, Ast-1 plants required a 200 µM dose for a similar response [65]. These observations led the authors to suggest that A10.1 plants are more sensitive to ABA signaling, which may explain their distinct physiological responses to salt stress.

In addition to phytohormone signaling, plants utilize active mechanisms to mitigate salt stress. Ion homeostasis plays a critical role in this process, as ion channels help expel Na^+^ and Cl^-^ back into the soil or sequester them into vacuoles to prevent cytoplasmic toxicity, while some species better adapted to saline environments have more specialized mechanisms, such as salt glands on their leaves [132,149]. In *S. viridis*, [145] demonstrated that vacuole expansion under salt stress can alter chloroplast structure, further underlining the importance of ion compartmentalization in stress responses.

One well-studied mechanism for salt tolerance in *S. viridis* involves the selective uptake and accumulation of Na^+^ and K^+^ by the roots [143,144]. Maintaining a proper Na^+^/K^+^ balance is crucial for cellular function and survival under salinity, as K^+^ is vital for osmotic regulation and proper nutritional homeostasis [150]. Ref. [143] investigated the potential of potassium supplementation to alleviate salt stress in *S. viridis*. Their results showed that plants treated with 5 mM KCl exhibited improved biomass accumulation and healthier root morphology when exposed to 150 mM NaCl, particularly in accessions A10.1 and ME034V [143].

Complementing these findings, [144] reported that potassium supply mitigates the effects of salinity on photochemical performance. KCl supplementation reduced chlorophyll fluorescence peaks in both untreated and salt-stressed plants. Notably, the performance index derived from chlorophyll fluorescence analysis improved significantly with KCl application [144]. However, KCl had limited effects on antioxidant enzyme activity, except for catalase, which showed increased activity in plants treated with 9 mM KCl. Additionally, electrolyte leakage—a marker of membrane damage associated with oxidative stress—declined following KCl supplementation [144]. These findings suggest that potassium improves stress resilience by stabilizing membrane integrity. Nevertheless, potassium supplementation failed to mitigate the adverse effects of severe salt stress on plant physiology and morphology [144] ), suggesting that potassium-mediated stress tolerance may be most effective under moderate salinity conditions. Further research is needed to clarify the mechanisms underpinning K^+^ homeostasis and its interplay with other stress mitigation strategies in *S. viridis*.

Beneath the morphophysiological responses to salt stress are numerous, complex molecular pathways activated in response to salinity. Changes in the transcriptional landscape triggered by abiotic stress often represent adaptive responses, with promising potential for biotechnological applications [67]. So far, most molecular studies have focused on *S. italica* and revealed that diverse gene families are responsive to salt stress. Ref. [58] investigated the role of lipoxygenases (LOX) in salt stress. Lipoxygenase enzymes play a role in the jasmonate (JA) pathways associated with salt stress, and JA is known to enhance salt tolerance in plants [151,152]. The study found *SiLOX2*, *SiLOX6*, *SiLOX8*, and *SiLOX9* upregulated under salt stress, with *SiLOX10* and *SiLOX11* further upregulated in the salt-tolerant cultivar QH2 [58]. Additionally, MADS-box genes were implicated in salt stress response. Ref. [69] reported significant induction of *SiMADS01* and *SiMADS51* among 10 genes previously associated with drought and salinity. Similarly, TCP transcription factors—targets of *miR319*, a known regulator of salt stress response [67,153]—were analyzed by [154]. Of the 22 TCP genes studied, *SiTCP2*, *SiTCP3*, *SiTCP4*, *SiTCP5*, and *SiTCP12* were repressed following salt stress.

Non-specific lipid transfer proteins (LTPs) have also been linked to salt stress tolerance. [155] characterized *SiLTP* in *S. italica*, demonstrating its induction during early salt stress followed by a decline with prolonged exposure. LTPs are small peptides able to transfer phospholipids between membranes and that have been reported to participate in vegetative and reproductive development, as well as in pathogen defense and abiotic stress response [155]. Interestingly, when tobacco plants overexpress *SiLTP*, they show higher germination and survival rates, and higher levels of accumulated proline and soluble sugars [155]. Notably, overexpression of *SiLTP* in *S. italica* improved root and shoot growth under 100 mM saline conditions, while RNAi knockdown lines displayed greater sensitivity [155].

The ABA signaling pathway plays a pivotal role in both physiological and molecular responses to salt stress [67]. Ref. [66] reported changes in the expression of ABA-responsive SnRKs and PP2Cs in *S. viridis* following 48 h of salt stress, with PP2Cs being particularly responsive. Notable genes such as *SnRK2.1, SnRK2.9, PP2C6, PP2C8*, and *PP2C7.1* were highly expressed in A10.1 plants. Interestingly, the Ast-1 accession exhibited significant induction of PYL genes, contrasting with their downregulation in A10.1 genotype [65].

Transcriptome analysis provides valuable insights into global transcriptional changes during salt stress. In the study by [145], salinity stress resulted in over 6000 differentially expressed genes (DEGs) in *S. viridis*, enriched for GO terms related to oxidation-reduction processes, carbohydrate metabolism, and hydrolase activity. Numerous transcription factors (TFs), including MYB, WRKY, ERF, HSF, and HD-ZIP, were prominently involved, with 109 TFs being overregulated and 58 repressed. MYB TFs were particularly abundant among DEGs [145].

Ref. [156] analyzed five families of C_4_ pathway genes in *S. italica*: carbonic anhydrase (CaH), phosphoenolpyruvate carboxylase (PEPC), pyruvate orthophosphate dikinase (PPDK), NADP-dependent malate dehydrogenase (MDH) and the NADP-dependent malic enzyme (NADP-ME). Among the numerous identified genes, only *SiαCaH1* and *SiPEPC2* were induced in the salt-tolerant cultivar IC-4, while *SiMDH8* was induced in the sensitive cultivar IC-41 [156]. Furthermore, overexpression of *SiNADP-ME5* in yeast enhanced salt tolerance, despite its lack of induction under salinity in plants [156]. Ref. [145] examined gene expression related to the C_4_ pathway, photorespiration, and oxidative stress responses in *S. viridis* compared to the halophyte Spartina alterniflora. Significant upregulation of PEP-CK1 and CAT2 occurred in *S. viridis*, along with genes encoding peroxidase superfamily protein and flavonol synthase/flavonone 3-hydroxylase. Conversely, genes related to oxidation-reduction processes such as ATLOX2, ACSF, CZSOD2, PORA, FAD8, and GAPB were strongly downregulated. Interestingly, the halophyte was much more stable in its transcript analysis in contrast to *S. viridis*, emphasizing the susceptibility of *S. viridis* to salt stress [145].

In a complementary study, [157] explored the transcriptional landscape of two *S. italica* cultivars (salt-tolerant and salt-sensitive) during seed germination. The tolerant cultivar exhibited 362 DEGs during seed imbibition (IM) and 1520 DEGs during radicle protrusion (RAP), whereas the sensitive cultivar showed 828 DEGs during IM and 3040 DEGs during RAP. Several transcription factors belonging to the ERF, bHLH, MYB, HD-ZIP and bZIP families are also responsive to salt stress [157]. DEGs were associated with ABA and gibberellic acid signaling, auxin and brassinosteroid biosynthesis, primary metabolism, and energy production, underscoring the critical role of metabolic activity during germination under stress. Genes related to photosynthesis, chloroplast development, and cell wall modification were upregulated, suggesting anticipatory autotrophic growth as a stress tolerance strategy [157]. Functional validation of two candidate genes, *SiDRM2* and *SiKOR1*, involved in hormonal regulation and associated with cell wall modification, respectively, revealed opposing effects: *SiDRM2* overexpression in Arabidopsis enhanced germination under salinity stress, while *SiKOR1* overexpression reduced germination, highlighting their regulatory roles in hormonal signaling and cell wall dynamics [157].

Similarly, [140] examined 104 *S. italica* accessions under salt stress and compared two representative genotypes: FM6 (tolerant) and FM90 (sensitive). The FM90 accession exhibited more extensive gene repression (1100 DEGs) and induction (800 DEGs) compared to the FM6 accession (345 repressed, 391 induced) [140]. Transcription factors such as AP2/ERF, HSF and WRKY were repressed in the sensitive genotype, while GATA genes were mostly induced. The tolerant cultivar showed significant induction of NAC TFs, which have been associated with stress resilience [140]. GO enrichment analysis revealed significant downregulation of photosynthesis-related genes in the sensitive genotype, suggesting impaired energy metabolism, whereas the tolerant accession maintained stability in photosynthetic pathways and displayed enrichment for ion transport processes [140]. Protein–protein interaction networks highlighted repressed ribosomal proteins as central hubs in the sensitive genotype, which the authors associated with slowing growth as a survival [140]. In contrast sugar metabolism hub genes dominated the tolerant genotype network, suggesting alternative energy production pathways to mitigate stress [140], similar to the observations on autotrophic growth made by [157].

## 6. Nutrient Deficiency

Plant mineral nutrition is a complex and multifaceted topic. Essential nutrients are classified as macro or micronutrients based on their required quantities and play critical roles in plant physiology [158] (Figure 3). Nutrient deficiencies can lead to drastic outcomes, compromising plant growth and yield. Despite the importance of mineral nutrition, responsesthe response of *Setaria* species to different types of nutritional stress remains underexplored. Existing studies primarily focus on nitrogen, phosphate, and potassium deficiencies, particularly in *S. italica*, which is known for its tolerance to low-fertility conditions [159,160].

### 6.1. Nitrogen Deficiency

Nitrogen (N) is indispensable for plant life, as it forms the basis of proteins, nucleic acids, vitamins, and other essential biomolecules [158]. Plants primarily absorb nitrogen in the form of nitrate, with ammonium as an alternative source [161]. Nitrogen deficiency manifests in diverse ways, including leaf chlorosis, necrosis, growth inhibition, reduced photosynthetic pigment levels, and a decline in amino acids and protein content [158]. In *S. viridis* and *S. italica*, nitrogen starvation causes fewer leaves to develop and leaf discoloration due to reduced chlorophyll and carotenoid levels [138,162]. Shoot and root growth are reduced, as well as shoot dry weight [160,162]. Root architecture changes in response to low nitrogen, developing shorter and less numerous lateral and crown roots [160]. Additionally, nitrogen stress compromises reproductive output, leading to pale, thin panicles and reduced seed production in *S. viridis* [138,163]. The nutritional quality of seeds is also affected, as seen in decreased folate content under nitrogen starvation in *S. italica* [163].

In addition to the morphological changes, *S. italica* suffers a significant drop in nitrogen content in both shoots [162] and roots [160]. The decrease in nitrogen content in roots is much sharper than in the shoot, indicating some degree of N mobilization to the shoot in response to low nitrogen [160]. Nitrogen starvation also triggers metabolic shifts: soluble proteins increase in roots while total amino acids decline, whereas both parameters drop in shoots [160,162]. Moreover, enzymes involved in nitrogen metabolism are differentially regulated; for example, glutamate synthase (GOGAT) activity increases, whereas glutamine synthetase (GS), nitrate reductase, and nitrite reductase activities are inhibited [162]. These changes indicate how nitrogen use efficiency (NUE) shifts in *S. italica* under low nitrogen stress. Ref. [160] have reported a threefold increase in NUE in the shoot, while roots displayed a twofold increase, suggesting the N mobilization to the shoot might be a strategy to cope with low nitrogen and supply the photosynthetic demand.

At the molecular level, nitrogen starvation downregulated chloroplast-related genes, including *SiPEPC*, which is critical for carbon assimilation during photosynthesis [162]. Genes involved in nitrogen transport and assimilation also exhibit dynamic responses. For instance, *SiNRT1.11* and *SiNRT1.12* are upregulated in shoots, while *SiNRT1.1*, *SiNRT2.1*, *SiNAR2.1*, and *SiAMT1.1* are induced in roots, enhancing nitrate and ammonium uptake and mobilization [160].

Comprehensive analyses of the NITRATE TRANSPORTER 1/PEPTIDE TRANSPORTER (NPF) family in *S. italica* have revealed its critical role in nitrogen transport during stress. Most NPF genes are induced under low nitrogen, particularly in shoots, with expression levels increasing over time [159]. Curiously, both *S. italica* and *S. viridis* have tandem duplications of NRT1.1B, unlike other crops, such as rice and sorghum. Moreover, both copies are highly expressed in vegetative tissues [159]. These findings suggest this duplication event might have played a role in the higher tolerance of *Setaria* species to low fertility. The nitrate-transporting ability of *SiNRT1.1* was confirmed by generating transgenic Arabidopsis, complementing the nrt1.1 mutant phenotype. Furthermore, a chlorate sensitivity assay [141] showed that transgenic plants had better nitrate absorption rates than wild type [159]. These results combined provide further evidence for the role of *SiNRT1.1* in low nitrogen tolerance.

In addition to nitrate transporters, other genes are responsive to low nitrogen and play an important role in the response of *S. italica* to nitrogen starvation. *SiATG8a* is a gene related to autophagy pathways and is responsive to nitrogen deficit [164,165]. Separate studies have demonstrated that overexpression of *SiATG8a* leads rice [165] and Arabidopsis [164] plants to have increased tolerance to low nitrogen. Interestingly, transgenic rice overexpressing *SiATG8a* had better survival rates, shoot dry weight and height compared to the wild type under stress, while the soluble protein content was significantly decreased [165]. These findings suggest that soluble peptides act as an important nitrogen reserve, and their rapid degradation plays a fundamental role in mitigating low nitrogen stress.

Additionally, transcription factors such as *SiMYB30* further mediate nitrogen starvation responses. *SiMYB30* is induced in *S. italica* after 24 h under low nitrogen [166]. Transgenic rice overexpressing *SiMYB30* showed a low nitrogen-tolerant phenotype, with increased fresh and dry weight, shoot height and root area compared to the wild type under low nitrogen. Nitrogen content in the seeds of transgenic plants also increased, indicating better NUE [166]. Interestingly, *NRT*1, *NIA*2 and *GOGAT*1 were all induced under low nitrogen in transgenic plants, while NPF2.4, NRT1.1B, e *GOGAT*2 were induced regardless of treatment [166]. Promoter analysis revealed all of these genes had MYB binding elements in the promoter region and luciferase assays demonstrated how *OsGOGAT*2 is directly regulated by *SiMYB30* [166].

### 6.2. Phosphorus Deficiency

Phosphorus (P) is another macronutrient essential for plant growth and development. Similarly to nitrogen, phosphorus is a fundamental component of nucleic acids and amino acids, as well as phospholipids and ATP molecules [158]. Phosphorus is primarily available in the soil as inorganic phosphates [158]. Phosphorus deficiency commonly leads to decreased growth, mainly due to reduced ATP synthase activity and impaired regeneration of NADPH. This is often accompanied by anthocyanin accumulation, leaf necrosis, and reduction in yield [158].

*S. italica* exhibits adaptive mechanisms under low phosphate conditions, developing an extensive root system to maximize phosphate acquisition. This contrasts with the reduced root growth typically observed in low nitrogen substrates. Lateral roots increase in number, length and density. Aerial growth, however, suffers a significant reduction [167]. Additionally, phosphate levels drop across the entire plant, and phosphate use efficiency (PUE) increases, particularly in the roots [167]. Phosphate deficiency induced hormonal changes in *S. italica*, with increased auxin and gibberellin levels. ABA levels rise drastically in roots but decrease significantly in shoots, highlighting a complex hormonal interplay in response to stress [167].

Phosphate starvation induces the expression of several phosphate transporter genes, such as *SiPHT1;1*, *SiPHT1;2* and *SiPHT1;4* [167,168]. Ref. [ [168]] analyzed the expression of the PHT1 family of phosphate transporters in 20 *S. italica* accessions under low phosphate. They observed that the most efficient genotypes showed positive correlations between the expression of *SiPHT1;1*, *SiPHT1;2*, *SiPHT1;3*, *SiPHT1;8*, and the plants’ phosphorus content [168]. These findings underscore the crucial role of PHT1 transporters in phosphate absorption and transport during phosphorus deficiency. However, their potential application in developing tolerant cultivars remains unexplored, presenting an avenue for future research.

Phosphate deficiency leads to an increase in free amino acid content in *S. italica*, suggesting heightened protein degradation activity as a potential adaptive response [167]. Supporting this, [169] conducted a complementary study on the role of *SiATG8a* during phosphate starvation. Consistent with earlier findings [164], *SiATG8a* is also induced by low phosphate, and transgenic wheat overexpressing *SiATG8a* showed a more tolerant phenotype under phosphate starvation [169]. Transgenic plants exhibited significant increases in spike number, grain yield, and phosphorus content in leaves and roots compared to the wild type [169]. Further analysis revealed that *SiATG8a* overexpression induced several phosphate transporters, including PHR1 and PT9 across the whole plant, PHR2 and PT3 in the shoot, and PAP10 and IPS1 in the roots [169]. These collective findings highlight a pivotal role for autophagy and protein degradation in the plant’s nutrient deficiency response. Additionally, they suggest that soluble proteins might act as crucial nutritional reserves during stress.

### 6.3. Potassium Deficiency

Alongside nitrogen and phosphorus, potassium (K) completes the trifecta of the major essential macronutrients for plant growth and development. Potassium is indispensable for meristematic growth, facilitating processes such as cell wall expansion and maintenance of cell turgor, as well as enzyme activation, pH regulation, and stomatal aperture control [158]. Potassium is absorbed by roots in its cationic form (K^+^) and its deficiency manifests in older leaves as chlorosis, curling, necrosis, and the development of shoots with shorter, weaker internodes. Additionally, potassium starvation adversely affects nitrogen availability, disrupting amino acid and protein synthesis [158]. Despite its importance, studies on potassium deprivation in the *Setaria* genus remain sparse, highlighting significant gaps in our understanding of *Setaria* mineral nutrition.

Potassium deficiency in *S. italica* leads to reductions in both shoot and root growth, culminating in diminished fresh and dry biomass [170,171]. A comprehensive study by [171] on various *S. italica* accessions identified Longgu 25 as particularly tolerant to low potassium conditions. Transcriptomic analysis revealed 1982 DEGs under low potassium, including transcriptional factors from families such as MYB, AP2, NAC, Homeobox, bHLH, WRKY and bZIP [171]. Notably, *SiMYB3* was highly induced by potassium starvation. Overexpression of *SiMYB3* in Arabidopsis enhanced tolerance to potassium deprivation, evidenced by increased fresh weight, longer primary roots, and a larger relative root area in comparison to the wild type [171]. These findings suggest *SiMYB3* promotes root elongation and low potassium tolerance, making it a promising candidate for crop improvement.

Furthermore, a recent study by [170] identified Yugu28 as another *S. italica* accession exhibiting high tolerance to low potassium. Over 4000 DEGs were identified under low potassium, including many transcriptional factors [170]. Consistent with the observations of Cao et al. (2019), MYB TFs were highly represented among the DEGs [170]. Most of the potassium transporters identified were upregulated by stress, while a specific ion-binding protein was repressed, suggesting their involvement in potassium homeostasis and stress response [170].

In addition to TFs and transporters, DEGs related to hormonal pathways were prominently represented in Yugu28. Auxin-associated growth regulators, including AUX/IAA genes, were particularly prevalent, alongside genes responsive to ethylene and gibberellic acid [170]. Intriguingly, *SiSnRK2.6*, a gene closely linked to the ABA signaling pathway, was also highly induced by the low potassium stress. Transgenic rice plants overexpressing *SiSnRK2.6* demonstrated enhanced tolerance, exhibiting larger and more robust root systems and taller shoots [170]. These findings highlight the critical role of hormonal regulation during potassium deficiency and underscore the ABA signaling pathway’s contribution to plant survival in potassium-depleted soils.

## 7. Heavy Metals

Anthropogenic activities, particularly the use of inorganic fertilizers and pesticides, are primary sources of heavy metal contamination in agricultural soils [172]. Industrial processes such as mining, smelting, and refining further exacerbate this issue by releasing heavy metals into the environment through effluents, aerosols, and leaching, thereby polluting soils, water bodies, and groundwater [173,174]. Plants cultivated in these contaminated substrates readily absorb heavy metals, initiating a cascade of toxic effects (Figure 3).

Heavy metals disrupt primary metabolism across various phenological stages, inducing a wide spectrum of physiological and metabolic alterations [175]. A primary phytotoxic effect is the inhibition of germination and root system development, often resulting from impaired cell division and elongation. Metals such as Hg, Cd, Co, Cu, Pb, and Zn are known to suppress germination, root elongation, and shoot growth [176,177,178], with root elongation inhibition being a particularly sensitive toxicity parameter. The order of toxicity often follows the stability of metal–organic complexes formed between cations and plant constituents [179].

Photosynthesis is a key target of heavy metal toxicity. While essential as cofactors, excess metals cause foliar chlorosis, linked to reduced chloroplast density and size [180]. Moreover, metal toxicity damages chloroplast ultrastructure and disrupts the finely tuned electron transfer chain and its coordination with carbon fixation [181,182]. This is compounded by a reduction in leaf area and number, an adaptive response to oxidative stress that also reflects stunted growth and lower nitrogen uptake [15]. Heavy metals also inhibit the biosynthesis and accumulation of photosynthetic pigments through enzymatic degradation [183].

At the molecular level, heavy metal accumulation induces structural changes in chloroplast membranes, lipids, and photosystems (PSI & PSII). This leads to impaired photosynthetic electron transport (PET), reduced chlorophyll synthesis, and the downregulation of carbon fixation [184]. The water-splitting complex on the PSII oxidizing side and ferredoxin-NADP^+^ reductase on the PSI reducing side are identified as particularly metal-sensitive sites [185]. The loss of chlorophyll, a commonly reported symptom, directly damages the photosynthetic apparatus [105]. Specific metals like cadmium (Cd) directly inhibit chlorophyll biosynthesis [186,187,188] and disrupt proper chloroplast development [189,190], ultimately suppressing net photosynthesis and impairing key enzymes like Rubisco [55].

Despite the widespread detrimental effects of heavy metal contamination, tolerance mechanisms and remediation strategies remain underexplored in C_4_ species, and studies on *S.*
*viridis* have been instrumental in advancing our understanding of heavy metal stress. Ref. [191] demonstrated that transgenic *S. viridis* plants overexpressing a Brachypodium distachyon MATE gene (*BdMATE*) exhibited enhanced aluminum tolerance. This was characterized by sustained root growth and aluminum exclusion from the root apex, a process linked to greater citrate exudation into the rhizosphere, suggesting metal chelation as the key tolerance mechanism.

Symbiotic relationships can also modulate metal stress responses. Ref. [192] found that inoculating *S. viridis* with the arbuscular mycorrhizal fungus *Funneliformis mosseae* mitigated cadmium (Cd) toxicity. The fungal symbiont induced growth by boosting antioxidant enzyme activity, eliminating reactive oxygen species, and maintaining greater plant biomass under stress. Furthermore, *S. viridis* shows promise in phytoremediation, with reports that the plant could tolerate and grow in soil amended with 40% coal gangue, with the amendment itself enhancing the plant’s resistance to metal and oxidative stress, underscoring its potential for restoring contaminated environments [193].

Studies on foxtail millet (*S. italica*) provide further insights into heavy metal stress in crops. Ref. [194] observed that Cd stress significantly disrupted morphophysiological parameters in millet, including reductions in shoot and root length, leaf number, panicle biomass, and chlorophyll content, with the most severe effects at 1.5 µM Cd. At the molecular level, [195] conducted a comprehensive analysis of the Natural Resistance-Associated Macrophage Protein (Nramp) genes in foxtail millet, identifying 12 *SiNramp* genes. The study revealed that these genes exhibited distinct, tissue-specific expression patterns in response to cadmium (Cd) stress.

The authors specifically linked *SiNramp6* and *SiNramp12* to different metal transport roles based on correlation analysis: *SiNramp12* expression in roots was upregulated under high Cd stress and showed a strong positive correlation with the accumulation of Cd, Fe, and Cu, suggesting a role in transporting these metals. In contrast, *SiNramp6* showed a unique expression pattern, and its transcript levels were positively correlated with the accumulation of essential nutrients like Ca, Zn, and Mg, indicating a potential role in the homeostasis of these elements, possibly in competition with toxic Cd uptake. This functional differentiation, alongside evidence of gene duplication and diverse protein structures, underscores the expansion and specialization of the *SiNramp* gene family in foxtail millet’s response to metal ion stress.

Remediation strategies have also been effectively tested in foxtail millet. Ref. [196] demonstrated that the application of biochar, especially that pyrolyzed at 500 °C (BC500), significantly attenuated Cd and Zn phytotoxicity. BC500 treatment reduced bioavailable metal content, promoted millet growth, mitigated oxidative stress, and lowered the translocation of Cd from roots to shoots, resulting in the highest plant biomass and photosynthetic rates. Moreover, gene expression studies revealed that the effects of copper and aluminum (in both ionic and nanoparticle forms) on foxtail millet seedlings are concentration dependent. Generally, low concentrations (e.g., 0.4 mg L^−1^) stimulated the growth rate and the expression of stress-related genes, such as ACT-1, CDPK and P5CS, while higher concentrations were inhibitory, highlighting a complex transcriptional response to nanometal stress [197].

Collectively, these studies demonstrate that key mechanisms behind heavy metal stress responses include the chelation and exclusion of metals through root exudates, symbiotic relationships that enhance antioxidant defense, and the orchestrated action of specific transporter gene families which manage the complex balance between essential nutrient uptake and toxic metal accumulation.

## 8. Combined Stimuli and Other Abiotic Stresses

Current climate change projections indicate that combined abiotic stresses, rather than isolated events, will become more frequent in the near future. Co-occurring stresses trigger multiple response pathways simultaneously and often result in different outcomes compared to isolated stresses [198,199]. Furthermore, combined abiotic stressors compromise plant defenses against biotic stressors, increasing vulnerability to pests and pathogens [200]. Despite the practical importance of understanding multi-stress scenarios, very few studies focus on *Setaria* species’ responses to combined stresses. In this section we discuss some works on co-occurring abiotic stresses in *S. viridis* and *S. italica* while highlighting this significant gap in knowledge.

The foundational study by [34] analyzed the genetic diversity within *S. viridis* to investigate the physiological, molecular and biochemical bases of tolerance to combined water and heat stress. Among the six evaluated accessions, Zha-1, A10.1 and Ula-1 emerged as stress-tolerant, while Ast-1, Aba-1 and Sha-1 were considered sensitive genotypes. Under combined heat and water stress, the sensitive accessions suffered a significantly greater biomass loss compared to individual stresses, highlighting the synergistic negative effect of these conditions. Meanwhile, the tolerant accessions maintained higher leaf water potential, photosynthetic and transpiration rates, and stomatal conductance under all stress conditions [34]. At the biochemical and molecular levels, stress resilience was underpinned by increased expression of aquaporins and ROS scavenging enzymes, moderate ABA accumulation, resulting in less severe stomatal closure, and enhanced metabolic stability resulting in milder declines in pigment content and total protein and sugar content. In contrast, the sensitive accessions exhibited inhibition of photosynthetic genes (e.g., PS1, RbcS, PEPC), downregulation of aquaporins and exaggerated ABA response, culminating in metabolic disarray, lipid peroxidation and growth inhibition [34].

A similar study by [201] dissects the response to combined drought and high-temperature stress in *S. italica* and compares the differences between the tolerant cultivar ISe-15 and the susceptible ISe-254. The tolerant cultivar exhibited higher ABA accumulation and a greater reduction in gibberellic acid content under stress; a response related to better water preservation. Additionally, metabolic profiling showed that ISe-15 accumulated more carbohydrates and protective compounds such as phenols and hydroxycinnamic acids, which play important roles in ROS scavenging and strengthening the cell wall. Meanwhile, the susceptible ISe-254 genotype showed higher levels of fatty acids, indicative of greater lipid peroxidation and oxidative damage. The study suggests tolerance in ISe-15 is associated with the pyrimidine metabolism pathway, which supports energy metabolism and cell wall biosynthesis under stress [201].

A field study investigating the interaction between nitrogen nutrition and drought stress demonstrated that nitrogen supplementation can significantly mitigate the adverse effects of water deficit in *S. italica* [201]. Under severe drought stress (85% soil water depletion) the H_2_O_2_ and MDA content were significantly increased, leading to a 52% grain yield reduction. The application of nitrogen fertilizer played a crucial protective role, enhancing chlorophyll and carotenoid, proline and phenolic compounds content, and reducing lipid peroxidation and ROS accumulation. This nitrogen-mediated stress mitigation increased yield by 16% under drought conditions [202]. In summary, this work establishes that nitrogen nutrition can be employed as an efficient alleviating strategy for drought-induced oxidative stress in foxtail millet by reinforcing its biochemical defenses.

Recent research has begun to elucidate the complex interplay between light quality and temperature in plant stress responses. The study by [78] not only analyzed the isolated responses of *S. viridis* to cold stress and end-of-day far-red (EOD-FR) light, but they also investigated the combined effect of both stresses. When chilling stress and EOD-FR overlap, there are 199 DEGs shared between both responses. These DEGs are associated with light signaling, trehalose metabolism, peroxidases and cold stress pathways even before the cold stress is applied [78]. These findings suggest that EOD-FR serves as a priming stimulus before steep drops in temperature. The expression of phytochrome PHYA and phytochrome-interacting factor PIF8 was significantly increased when EOD-FR and cold stresses were combined, supporting the proposed model where EOD-FR improves cold tolerance via a PHYA-jasmonic acid-C-repeat DREB factors (CBF) regulatory pathway [78]. Additionally, the study builds regulatory networks and highlights genes related to calcium signaling, MAPK cascades, trehalose metabolism genes and transcription factors such as BBX2. BBX2 expression was highly induced by chilling and further increased when EOD-FR was introduced, suggesting BBX2 is a potential critical regulator at the core of far-red light and cold signaling [78]. In conclusion, this study demonstrates that EOD-FR light is not merely a developmental signal but a key priming agent that rewires the transcriptional network to enhance chilling tolerance in *S. viridis*.

In addition to the broadly studied abiotic stresses covered in this review, there are limited studies on the response of *Setaria* to lodging and waterlogging. Lodging is the mechanical displacement of plant stems, frequently caused by strong winds and/or rain, leading to significant yield losses. The structural strength of the internodes is a key factor in lodging resistance, and a study in *S. italica* showed that the fourth and fifth internodes are significantly weaker and prone to breaking than the third internode [203]. Although stem strength is genetically determined, it is also strongly influenced by agricultural practices and decreases linearly with increased plant density [203]. In a complementary study, [[204]investigated the yield losses caused by stem lodging in *S. italica* and revealed that even transient lodging causes losses between 19.3% in accession Jigu 31% and 25% in Chenggu 13. Permanent lodging until maturity resulted in far more severe losses reaching up to 51.1% in accession Chenggu 13. Interestingly, the adverse effects were stage-dependent, with the grain filling stage identified as the most sensitive period [204]. The most critical factors affected by lodging are panicle weight and grain weight per panicle, resulting in significant yield losses, suggesting the primary cause of yield loss is a reduction in grain number and individual grain filling [204].

Waterlogging is characterized by the flooding of the soil and roots, while submergence encompasses the aerial parts of the plant. The sole study on the effects of waterlogging in *S. italica* by [205] reveals the marked susceptibility of foxtail millet to this type of stress [205]. Significant yield losses are observed under sustained waterlogging, and the timing of the stress is critical: waterlogging during the vegetative phase results in more severe damage than post-heading (reproductive phase). The primary cause of yield loss is a reduction in the number of filled grains per panicle rather than a change in panicle number or individual grain weight. These losses are attributed to declining leaf water potential, likely due to root dysfunction, decreased photosynthetic rate, linked to lower stomatal conductance, and overall reduced biomass accumulation [205]. Ref. [205] propose that the core reason behind this susceptibility is in the root system’s failure to adapt to oxygen-deficient conditions, highlighted by the lack of lysigenous aerenchyma to enable internal oxygen transport from shoots to roots. *S. italica* seemingly has limited resources to cope with waterlogging, which underscores the necessity for further studies into this specific stress, not only in the foxtail millet, but also in its wild relative *S. viridis*.

While significant progress has been made in understanding the physiological basis of lodging and waterlogging, further studies will be essential to unravel the genetic and molecular basis of these two stresses, which is a prerequisite for developing resilient millet varieties capable of withstanding the unpredictable precipitation patterns of a changing climate. To facilitate comparison across the different stress responses, a summary table is provided below (Table 1), highlighting the key morphological, physiological, and molecular effects observed in *S. viridis* and *S. italica*, alongside the corresponding references for each stress type.

## 9. Conclusions and Future Perspectives

Drought, temperature extremes, high luminosity, saline soil, nutrient deficiencies and heavy metal contamination represent major constraints on agricultural production, often reducing yield and causing significant economic losses. In the context of climate change, research into plant resilience to these pressures has become increasingly urgent. The crop–wild relative pair *S. italica* and *S. viridis* constitute a powerful comparative system for dissecting the genetic basis of abiotic stress tolerance in C_4_ grasses. Despite high genome synteny, they differ markedly in evolutionary history: *S. viridis* retains the broad allelic diversity and stress-adaptive plasticity of a widespread wild weed, whereas *S. italica* has undergone approximately 7900 years of domestication and modern breeding; processes that typically impose genetic bottlenecks, relax selection on stress-responsive pathways, and favor alleles for yield and uniformity under managed conditions [21,206,207]. This contrast provides a robust framework for identifying candidate genes and regulatory variants that enhance resilience and can be deployed in molecular breeding across other C_4_ species like maize and sorghum [208,209].

Studies leveraging this model have revealed a convergent strategy for abiotic stress resilience, centered on protecting photosynthesis and optimizing resource allocation. Across various stresses, plants prioritize defense of the photosynthetic apparatus through mechanisms like stomatal regulation, antioxidant enhancement, and energy dissipation. Simultaneously, precise management of ions and nutrients, whether excluding toxic Na^+^ or Al^3+^, foraging for phosphorus, or remobilizing nitrogen via autophagy, demonstrates a fundamental principle of maintaining metabolic homeostasis. For crops like *S. italica*, these adaptations not only ensure survival but also improve yield and grain quality. Underpinning these physiological adaptations is a conserved molecular toolkit. Signaling pathways, particularly those involving ABA, and key transcription factor families like NAC and MYB, orchestrate stress-responsive transcriptional reprogramming. The functional diversification of nutrient and metal transporter genes provides the genetic basis for efficient resource management. The contrasting evolutionary trajectories of *S. viridis* and *S. italica* have likely shaped the structure, diversity, and regulation of these very genes, including transcription factors, hormone signaling components, and cis-regulatory elements. Therefore, comparative analyses of sequence diversity, promoter architecture, and expression plasticity can reveal loci that were lost, attenuated, or rerouted during domestication [208,209].

Collectively, the findings reviewed here showcase how the *Setaria* system functions as an efficient discovery platform for stress-tolerance determinants with high translational potential. Looking forward, research should focus on underexplored stresses, pinpoint key regulatory nodes within the stress response networks, and continue to leverage the extensive genetic diversity of the *Setaria* genus to identify further traits of agricultural importance for climate-resilient agriculture.

## Figures and Tables

**Figure 1 plants-14-03710-f001:**
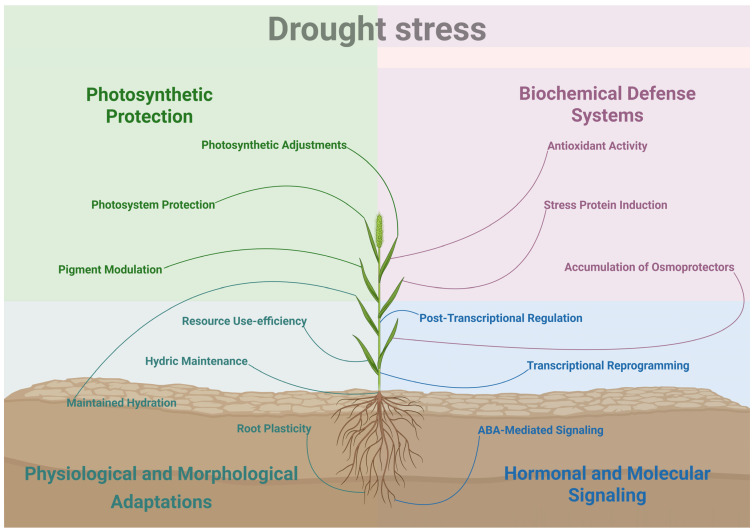
Overview of multi-level drought stress responses in *Setaria* spp. Water preservation is achieved via morpho-physiological adaptations such as stomatal closure, leaf rolling, altered root architecture and improved water use efficiency. Photoinhibitory damage is mitigated by increased non-photochemical quenching (NPQ) and adjustments to the photosynthetic machinery. Biochemical defense systems to protect organelles and cellular structures involve the accumulation of osmoprotectants and osmolytes such as proline, and the activation of antioxidant enzymes (e.g., SOD, CAT, POD). Finally, molecular responses are orchestrated by ABA signaling and transcription factor activation (e.g., NAC, DREB, MYB), leading to the regulation of protective genes (e.g., LEA, HSPs, Aquaporins).

**Figure 2 plants-14-03710-f002:**
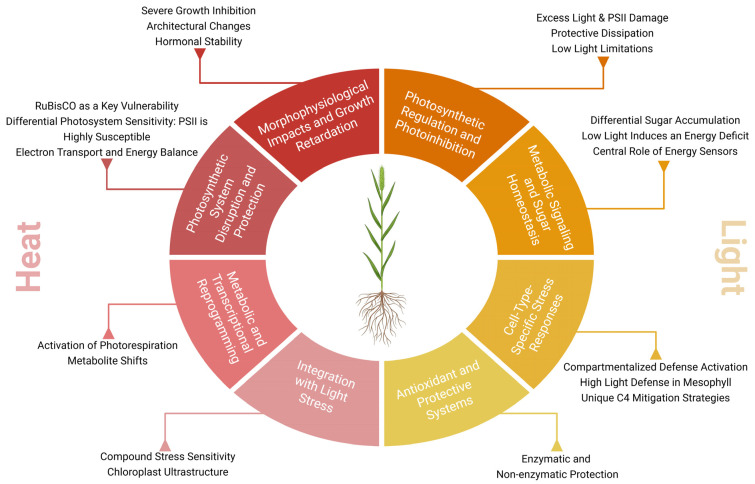
Integrated responses to elevated temperature and irradiance in *Setaria* spp. Heat and light stress usually co-occur and thus elicit similar responses such as photoprotection mechanisms, antioxidant enzymes and metabolic shifts. Heat stress results in reduced stomatal conductance and increased photorespiration and cyclic electron flow. Excessive light intensity directly modulates sugar-signaling pathways (e.g., SnRK1, HXK), carbon allocation and ROS-scavenging.

**Figure 3 plants-14-03710-f003:**
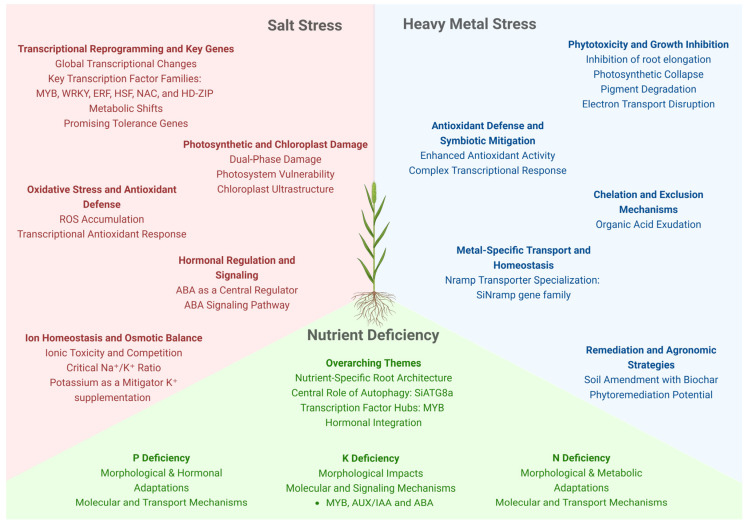
*Setaria* spp. stress responses to salinity, nutrient deficiency, and heavy metal toxicity. Under saline soils, ion homeostasis is central; Na^+^ is excluded through the roots or sequestered into the vacuole, while K^+^ uptake is prioritized. Several TF families (MYB, NAC, WRKY, bZIP) respond to salt stress and modulate the molecular defenses against it. Root architectural changes are key under nutrient starvation: while low nitrogen reduces root growth, phosphorus deficiency promotes lateral root proliferation and enhanced foraging. Autophagy pathways (e.g., SiATG8a) are depled under nitrogen or phosphorus scarcity, and soluble proteins act as nutritional reserves. Moreover, integrated hormonal and transcriptional regulation are critical for N, P and K deficiencies. Heavy metal stress is mitigated by ion chelation and compartmentalization/exclusion via metal transporters (e.g., Nramp). Remediation strategies such as soil amendment with biochar and phytoremediation strategies show promise in contaminated soils.

**Table 1 plants-14-03710-t001:** Comparative table presenting the main morphological, physiological, and molecular responses of the two species under different abiotic stresses: light, heavy metals, nutrient imbalance, salinity, drought, and high temperature. Highlighting the response mechanisms and the effects observed for each type of stress.

Drought Stress		
	Effect	Reference
**Growth and** **developmental** **effects**	Shoot dry mass decreases, leaf and tiller emergence slow down, increasing root length and surface area. Water status effects: leaf water potential and relative water content decline, wilting, leaf rolling and bleaching, reducing transpiration	[33,34,38,41,42,43,44,45,47,48,49,50]
**Photosynthesis &** **Biochemical** **effects**	Lower photosynthetic assimilation, degradation of pigments, loss of photosystem II integrity, reduced photochemical efficiency (qP, ΦPSII), overproduction of reactive oxygen species, damage cellular components, activation of antioxidant defenses and accumulation of osmolytes (proline)	[[19],[33],[34],[38],[41],[42],[45],[46],[47],[48],[49],[50],[51],[52],[53],[54],[55],[56],[59],[60],[61],[62],[63],[65],[66],[68],[69],[70][72][71],[73],[74]]
**Molecular effects**	Modulation of stress-responsive genes, members of PP2C*s*, SnRK2*s*, bZIP*s* families and metabolic enzymes (e.g., PGM)	[90,91]
**Stress temperature**		
	**Effect**	**Reference**
**Growth and** **Developmental** **effects**	Dwarfism and atrophy, leaf narrowing and total biomass reduction	[[79],[80][83][82],[84],[85],[86]]
**Photosynthesis &** **Biochemical** **effects**	Reduced photosynthetic efficiency, reduced activation of RuBisCO, reduction in chlorophyll biosynthesis, downregulation of PSII and cytochrome b6f and accumulation of osmoprotective sugars	[65,79,87,88,90,94,95,96,97,98,107,108,109,110]
**Molecular effects**	Repression of starch synthesis genes, upregulation of SGA*T*, SHM*T*, HP*R* and downregulation of the GLYK protein	[82,83,108,111,112,113,114,115,116]
**Saline Stress**		
	**Effect**	**Reference**
**Growth and developmental effects**	Delayed germination and reduced germination rates, Root emergence reduced, Leaf curling, burnt margins, increased leaf thickness, and reduced biomass accumulation.	[138,139,140,141,143,144]
**Photosynthesis &** **Biochemical** **effects**	Stomatal closure, reduced photosynthesis and ROS (reactive oxygen species) accumulation.	[65,132,139,142,143,11,145]
**Molecular effects**	*LOX* gene upregulation	[58]
**Nutritional Stress**		
	**Effect**	**Reference**
**Growth and** **developmental** **effects**	Delayed germination and reduced germination rates, reduced shoot and root growth, decreased shoot dry weight, shorter and fewer lateral and crown roots, reduction in leaf area, leaf chlorosis, necrosis, pale and thin panicles, reduced growth, leaf curling, burnt margins, chlorosis, increased leaf thickness, chlorosis, curling, necrosis in older leaves, shorter internodes, reduced biomass accumulation/diminished fresh and dry biomass, reduced aerial growth and increased lateral-root length, density, and number	[138,158,160,162,163,167,170,171]
**Photosynthesis &** **Biochemical** **effects**	Decline in chlorophyll and carotenoid levels, reduction in nitrogen content in shoots and roots; mobilization to shoots, decline in amino acids and protein content in shoots; soluble protein increase in roots, reduced photosynthesis and stomatal dysfunction and turgor loss	[138,158,160,162]
**Molecular effects**	Upregulation of nutrient-transport related genes, regulation by transcription factors (e.g., *SiMYB30*), upregulation of potassium transporters; repression of certain ion-binding genes (ion homeostasis) and induction of hormonal and signaling pathways (AUX/IAA, ethylene, ABA, gibberellin pathway genes)	[159,160,166,169,170,171]
**Light Stress**		
	**Effect**	**Reference**
**Growth and** **developmental** **effects**	Reduced growth, turgor and photosynthetic capacity in low light and change in daily photosynthesis pattern (curve from double peak to single peak)	[130]
**Photosynthesis &** **Biochemical** **effects**	Photoinhibition with functional failure of PSII, damage to PSII reaction centers, excessive generation of ROS damaging PSI and PSII, reduction in the maximum quantum efficiency of PSII, drop in CO_2_ assimilation rate, stomatal conductance, and water-use efficiency, alteration of fluorescence and dissipation of energy as heat (non-photochemical energy dissipation/photoprotection), accumulation of sugars under high light without suppression of photosynthesis and Activation of antioxidant mechanisms	[15,118,119,120,121,122,123,124,127,128,129,130,131]
**Molecular effects**	Upregulation of antioxidant genes and heat-shock proteins (HSPs) in mesophyll cells under high light, downregulation of antioxidant genes in bundle sheath under high light and reduced mitochondrial activity (sub-optimal light)	[15,120,121,123,124,130]
**Heavy-Metal Stress**		
	**Effect**	**Reference**
**Growth and** **developmental** **effects**	Inhibition of germination, reduction in root elongation, leaf area, and shoot growth, reduced biomass accumulation (fresh/dry weight) and pale or thin panicles and leaf chlorosis	[15,176,177,178]
**Photosynthesis &** **Biochemical** **effects**	Leaf chlorosis (loss of green pigment), damage to chloroplasts and damage to the photosynthetic electron transport (PET) chain, structural changes in chloroplast membranes and in photosystems PSI/PSII, reduction in photosynthetic pigments (chlorophyll, carotenoids), inhibition of chlorophyll biosynthesis, change in chloroplast development, inhibition of key enzymes involved in carbon fixation (e.g., Rubisco) and reduction in net photosynthesis/photosynthetic rate	[55,181,182,183]
**Molecular effects**	Expression of stress-responsive genes (e.g., *ACT-1, CDPK, P5CS*) and upregulation of detoxification and defense pathways (e.g., chelation, antioxidative responses)	[55,184,186,187,188,189,190]

## Data Availability

All data generated and analyzed during this study are included in this published article.

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
