# Peer review of "Harnessing Setaria as a Model for C4 Plant Adaptation to Abiotic Stress"

_plants, 2025, doi:10.3390/plants14243710_

Round 1

Reviewer 1 Report

Comments and Suggestions for Authors

The authors review harnessing of Setaria as a model for C4 plant adaptation to abiotic stresses. Overall, by citing numerous relevant papers, they provide a well-organized summary of individual studies on abiotic stress tolerance. I believe it is publishable, but as a minor point, the subscript for the 4 in C4 is inconsistent. Please review and correct this point.

Author Response

Review Report (Reviewer 1)

Comments 1: The authors review harnessing of Setaria as a model for C4 plant adaptation to abiotic stresses. Overall, by citing numerous relevant papers, they provide a well-organized summary of individual studies on abiotic stress tolerance. I believe it is publishable, but as a minor point, the subscript for the 4 in C4 is inconsistent. Please review and correct this point.

Response: Thank you for the suggestion. The “4” in “Câ‚„” has now been formatted as a subscript throughout the manuscript.

Reviewer 2 Report

Comments and Suggestions for Authors

Juan Gomes et al. present a comprehensive review that systematically explores the use of Setaria as a C4 plant model to dissect the physiological, biochemical, and molecular mechanisms underlying responses to abiotic stresses. At present, foxtail millet shows great potential in multiple applications, including food and nutrition, health products, animal feed, and biofuel production, while its wild relative, green foxtail, possesses rich genetic diversity and considerable research value. The manuscript summarizes the morphological, physiological, transcriptomic, and metabolic responses of Setaria crops under environmental conditions such as drought, extreme temperatures, high irradiance, salinity, nutrient deficiency, and heavy metal toxicity, which is important and will attract the attention and interest of relevant readers. I have several suggestions as follow.

  1. In addition to summarizing the responses of Setaria to various abiotic stresses, I recommend that the “Conclusions” or “Perspectives” section include a brief discussion of how abiotic stresses influence grain quality and nutritional components in foxtail millet, in order to strengthen the link between the review and the practical utilization value of this crop. For example:

Amoah, J.N., Adu-Gyamfi, M.O. & Kwarteng, A.O. Effect of drought acclimation on antioxidant system and polyphenolic content of Foxtail Millet (Setaria italica L.). Physiol Mol Biol Plants 29, 1577–1589 (2023). https://doi.org/10.1007/s12298-023-01366-w

  1. Given that multiple abiotic stresses often occur simultaneously under natural field conditions, I suggest that the authors provide a brief overview of combined stresses, or even a separate subsection devoted to this topic. A concise summary of current progress and existing knowledge gaps regarding Setaria under combined stress scenarios would enhance the real-world relevance and reference value of the review.

  1. The manuscript discusses the responses of S. italica and S. viridis to different abiotic stresses in several places, but the relevant information is scattered across various paragraphs. I recommend adding a summary table that lists, side by side, the main morphological, physiological, and molecular responses of the two species under major abiotic stresses (e.g., drought, salinity, high temperature), so that readers can more easily perform cross-species comparisons.
  2. The manuscript focus on Setaria C4 model and its stress tolerance, but a few high quality research related to this area are missing, I think these high quality research should be included and discussed:

Zhou Z, Zhang L, Wang Y, Zhang Y, Jia H, Zhi H, Jia G, Han Y, Diao X, Tang S. Multi-omics analysis of ubiquitin E2 genes in Setaria: evidence for the roles of E2 genes in various aspects of plant development, stress tolerance, and domestication. Plant J. 2025 Sep;123(5):e70473. doi: 10.1111/tpj.70473. PMID: 40944497.

Zhang P, Sharwood RE, Carroll A, Estavillo GM, von Caemmerer S, Furbank RT. Systems analysis of long-term heat stress responses in the C4 grass Setaria viridis. Plant Cell. 2025 Apr 2;37(4):koaf005. doi: 10.1093/plcell/koaf005. PMID: 39778116; PMCID: PMC11964294.

  1. At the structural level, certain sections could be further subdivided and refined. For example, in Section 6 on nutrient-related stresses, it would be clearer to separate nitrogen, phosphorus, and potassium into subsections 6.1, 6.2, and 6.3, respectively, so as to improve the hierarchy and readability of the text.

Minor issues:

  1. Please carefully proofread the entire manuscript for typographical and spelling errors. For instance:

   “In S. italica aquaporin genes are frequently up-regulated by drought, enhancing water uptake and transport [43,63]; in S. viridis drought-sensitive accessions exhibited lowest expression of aquaporins PIP-1, PIP1-2 and PIP2-1 compared to to tolerant accessions [31].”

In this sentence, the word “to” is duplicated at the end and should be corrected.

  1. Please pay attention to consistency of notation and formatting throughout the manuscript, especially for physicochemical symbols and abbreviations. For example, the notations CO2 vs CO2 and C4 vs C4 appear in different formats in the text; these should be standardized according to the journal’s style requirements.

Author Response

Juan Gomes et al. present a comprehensive review that systematically explores the use of Setaria as a C4 plant model to dissect the physiological, biochemical, and molecular mechanisms underlying responses to abiotic stresses. At present, foxtail millet shows great potential in multiple applications, including food and nutrition, health products, animal feed, and biofuel production, while its wild relative, green foxtail, possesses rich genetic diversity and considerable research value. The manuscript summarizes the morphological, physiological, transcriptomic, and metabolic responses of Setaria crops under environmental conditions such as drought, extreme temperatures, high irradiance, salinity, nutrient deficiency, and heavy metal toxicity, which is important and will attract the attention and interest of relevant readers. I have several suggestions as follow.

Comments 1: In addition to summarizing the responses of Setaria to various abiotic stresses, I recommend that the “Conclusions” or “Perspectives” section include a brief discussion of how abiotic stresses influence grain quality and nutritional components in foxtail millet, in order to strengthen the link between the review and the practical utilization value of this crop. For example:

Amoah, J.N., Adu-Gyamfi, M.O. & Kwarteng, A.O. Effect of drought acclimation on antioxidant system and polyphenolic content of Foxtail Millet (Setaria italica L.). Physiol Mol Biol Plants 29, 1577–1589 (2023). https://doi.org/10.1007/s12298-023-01366-w

R: We thank the reviewer for this suggestion. Although only one study to date has evaluated grain quality under stress in foxtail millet, we have added a brief discussion on this topic in the Conclusions section (lines 296–301), and we now cite the suggested reference (Amoah et al., 2023).

Comments 2: Given that multiple abiotic stresses often occur simultaneously under natural field conditions, I suggest that the authors provide a brief overview of combined stresses, or even a separate subsection devoted to this topic. A concise summary of current progress and existing knowledge gaps regarding Setaria under combined stress scenarios would enhance the real-world relevance and reference value of the review.

R: In response to this reviewer suggestion, we have added a new section entitled “Combined Stimuli and Other Abiotic Stresses” (beginning at line 1183). In this section, we provide a concise overview of the very limited literature on combined stress scenarios in Setaria viridis and S. italica. Additionally, we include under‑represented stress types (e.g., lodging and waterlogging), and we explicitly point out current knowledge gaps

Comments 3: The manuscript discusses the responses of S. italica and S. viridis to different abiotic stresses in several places, but the relevant information is scattered across various paragraphs. I recommend adding a summary table that lists, side by side, the main morphological, physiological, and molecular responses of the two species under major abiotic stresses (e.g., drought, salinity, high temperature), so that readers can more easily perform cross-species comparisons.

R: As suggested, we have included a summary table in the manuscript that compares the main morphological, physiological, and molecular responses of S. italica and S. viridis under major abiotic stresses (line 1255).

Comments 4: The manuscript focus on Setaria C4 model and its stress tolerance, but a few high quality research related to this area are missing, I think these high quality research should be included and discussed:

Zhou Z, Zhang L, Wang Y, Zhang Y, Jia H, Zhi H, Jia G, Han Y, Diao X, Tang S. Multi-omics analysis of ubiquitin E2 genes in Setaria: evidence for the roles of E2 genes in various aspects of plant development, stress tolerance, and domestication. Plant J. 2025 Sep;123(5):e70473. doi: 10.1111/tpj.70473. PMID: 40944497. 454-462

Zhang P, Sharwood RE, Carroll A, Estavillo GM, von Caemmerer S, Furbank RT. Systems analysis of long-term heat stress responses in the C4 grass Setaria viridis. Plant Cell. 2025 Apr 2;37(4):koaf005. doi: 10.1093/plcell/koaf005. PMID: 39778116; PMCID: PMC11964294. - 498-503 and 585 – 600

R: Following the reviewer’s suggestion, we have now incorporated discussion of both references in the appropriate sections of the revised manuscript (see lines 454–462 for Zhou et al., 2025; and lines 498–503 and 585–600 for Zhang et al., 2025)

Comments 5: At the structural level, certain sections could be further subdivided and refined. For example, in Section 6 on nutrient-related stresses, it would be clearer to separate nitrogen, phosphorus, and potassium into subsections 6.1, 6.2, and 6.3, respectively, so as to improve the hierarchy and readability of the text.

R: We agree with the reviewer and have subdivided the “Nutrient Deficiency” section into subsections (e.g., 6.1 Nitrogen, 6.2 Phosphorus, 6.3 Potassium) to improve readability and structure.

Minor issues:

Comments 6: Please carefully proofread the entire manuscript for typographical and spelling errors. For instance:

   “In S. italica aquaporin genes are frequently up-regulated by drought, enhancing water uptake and transport [43,63]; in S. viridis drought-sensitive accessions exhibited lowest expression of aquaporins PIP-1, PIP1-2 and PIP2-1 compared to to tolerant accessions [31].”

In this sentence, the word “to” is duplicated at the end and should be corrected.

R: The duplicated word “to” has been removed, and the manuscript has been thoroughly copy‑edited for grammar, spelling, and typographical errors.

Comments 7: Please pay attention to consistency of notation and formatting throughout the manuscript, especially for physicochemical symbols and abbreviations. For example, the notations CO2 vs CO2 and C4 vs C4 appear in different formats in the text; these should be standardized according to the journal’s style requirements.

R: In accordance with the reviewer’s suggestion, we have formatted “Câ‚„” and “COâ‚‚” correctly (subscript for the numerals) throughout the manuscript. We also conducted a comprehensive check to ensure consistency of all physicochemical symbols and abbreviations.

Reviewer 3 Report

Comments and Suggestions for Authors

The paper presents a comprehensive review of Setaria species in relation to plant adaptation to abiotic stress. This information is valuable for C4 crop species, as Setaria serves as a useful model. Overall, the paper has clear merit for publication in a journal ‘Plants’.

The authors mention that the two Setaria species form a unique sister pair, with Setaria italica being an important cultivated crop and Setaria viridis a globally distributed invasive weed. By comparing the physiological mechanisms underlying abiotic stress tolerance in both species, the reviewer believes that valuable insights could be gained for improving major C4 crops such as maize. However, this point is not explicitly discussed in the manuscript, and the reviewer recommends adding it.

Regarding temperature responses, low-temperature stress is also increasingly important due to climate change. In addition to drought tolerance, issues such as flooding should also be considered. If relevant findings exist, the reviewer suggests including them as well.

L103: It would be better to include specific numerical data regarding genome size and other related factors.

Author Response

The paper presents a comprehensive review of Setaria species in relation to plant adaptation to abiotic stress. This information is valuable for C4 crop species, as Setaria serves as a useful model. Overall, the paper has clear merit for publication in a journal ‘Plants’.

Comments 1: The authors mention that the two Setaria species form a unique sister pair, with Setaria italica being an important cultivated crop and Setaria viridis a globally distributed invasive weed. By comparing the physiological mechanisms underlying abiotic stress tolerance in both species, the reviewer believes that valuable insights could be gained for improving major C4 crops such as maize. However, this point is not explicitly discussed in the manuscript, and the reviewer recommends adding it.

R: We have added a new paragraph in the Discussion section (lines 1288–1325) to explicitly discuss how comparative physiological insights from Setaria could be translated to improving stress tolerance in major Câ‚„ crops such as maize.

Comments 2: Regarding temperature responses, low-temperature stress is also increasingly important due to climate change. In addition to drought tolerance, issues such as flooding should also be considered. If relevant findings exist, the reviewer suggests including them as well.

R: In line with the reviewer’s recommendation, we have extended the “Heat stress” section to “Extreme Temperatures” to include both heat and low‑temperature stress (starting at line 464). Additionally, we incorporated a brief discussion of flooding / waterlogging and lodging in the new “Combined Stimuli and Other Abiotic Stresses” section (from line 1183).

Comments 3: L103: It would be better to include specific numerical data regarding genome size and other related factors.

R: We have added numerical information regarding genome size, life cycle, and plant height for S. viridis and S. italica in the Introduction, as requested (see 7th paragraph, lines 101–114).
